# KV shifting attention enhances language modeling

**Mingyu Xu** [1]  **Bingning Wang** [1]  **Weipeng Chen** [1]

## Abstract

Current large language models (LLMs) predominantly rely on decode-only transformer architectures, which exhibit exceptional in-context learning (ICL) capabilities. It is widely acknowledged that the cornerstone of their ICL ability lies in the induction heads mechanism, which necessitates at least two layers of attention. To more effectively harness the model's induction capabilities, we revisit the induction heads mechanism and provide theoretical proof that KV shifting attention reduces the model's dependency on the depth and width of the induction heads mechanism. Our experimental results confirm that KV shifting attention enhances the learning of induction heads and improves language modeling performance. This leads to superior performance or accelerated convergence, spanning from toy models to pretrained models with over 10 billion parameters.

## 1. Introduction

Transformer-based Large language models (Vaswani, 2017) have demonstrated remarkable capabilities in in-context learning (ICL), largely attributed to the underlying mechanisms of induction heads (Elhage et al., 2021; Olsson et al., 2022). These mechanisms enable the models to identify and leverage repeating patterns, which is crucial in ICL (Song et al., 2024; Crosbie & Shutova, 2024) and multi-step reasoning (Sanford et al., 2024b).

Although there are many works based on transformer for analysis of induction heads of LLMs, there are few works that utilize analysis of induction heads to modify transformers to enhance its ability to learn induction heads. A better learning induction head may potentially contribute to the in-context learning or reasoning ability.

For the goal, we revisit the induction head mechanism and

analyze the required depth and width for induction heads. Based on the perspective, we propose KV shifting attention designed to simplify and enhance the induction process. By decoupling keys and values in the attention mechanism, KV shifting attention reduces the structural requirements for depth and width, enabling single-layer transformers to effectively perform induction.

Through theoretical analysis and empirical validation, we demonstrate that KV Shifting attention achieves comparable or superior performance to Vanilla attention. Moreover, its bias towards learning induction leads to more efficient and effective language modeling across diverse scales, from toy models to pre-trained models with billions of parameters. In summary, our contributions are as follows:

- To accelerates learning ability for induction heads, we revisit the mechanism of it, and analyze that KV shifting attention can effectively represent induction heads with lower depth and width, and learn induction heads from induction data.

- We apply KV shifting attention from toy model to large language modeling pre-training and demonstrate its effectiveness.

We arrange the paper in the following order. We introduces our motivation and methods in Section 2, analyzes the KV shifting attention in Section 3 , introduces our experiments on a large language model in Section 4, discuss, introduce relevant work, and summarize in the last few sections.

## 2. Method

### 2.1. Motivation

**Induction heads**   Our story begins with induction heads. Induction heads mechanism (Elhage et al., 2021; Olsson et al., 2022) is a circuit whose function is to find the latest same previous instances of the current token (call it A) and the token that came after it (call it B), then use B as current token's next token prediction. (e.g. forming the sequence [A][B]... [A] $\rightarrow$ [B]). It is generally believed that the implementation of the induction heads mechanism often **requires two heads belonging to different layers**. For a detailed proof that **a layer of transformer cannot**

---

[1]Baichuan-inc, China. Correspondence to: Bingning Wang <god@bingning.wang>.

*Proceedings of the $42^{st}$ International Conference on Machine Learning*, Vancouver, Canada. PMLR 267, 2025. Copyright 2025 by the author(s).

implement induction heads, readers can refer to Sanford et al. (2024a). So, can we make slight adjustments to the attention so that a single layer of attention can achieve the mechanism of induction heads?

**Virtual attention heads**   Another interesting perspective proposed in the article (Elhage et al., 2021) is the concept of virtual attention heads. It demonstrates the cooperation among attentions in different layers. For simplicity, we consider the attention of only two adjacent layers and ignore the presence of residual connections and MLP. The formal expression is:

$$X_1 = A^{h1}X_0 W_{ov}^{h1}, X_2 = A^{h2}X_1 W_{ov}^{h2}, \qquad (1)$$

where $X_0 \in R^{N \times D}$ is the input, $X_l \in R^{N \times D}$ is the output of $l^{th}$ layers, $A^{hl} \in R^{N \times N}$ is the attention weights of $l^{th}$ layer, $W_{ov}^{hl} = W_v^{hi}W_o^{hl} \in R^{D \times D}$, $W_v^{hl} \in R^{D \times D}$ is the $v$ projection, $W_o^{hl} \in R^{D \times D}$ is the $o$ projection of $l^{th}$ layer, $l \in 1, 2$, $n \in R$ is context length, $d$ is the dimension of hidden states. Then virtual attention heads is:

$$X_2 = A^{h2}X_1 W_{ov}^{h2} = (A^{h2}A^{h1})X_0(W_{ov}^{h1}W_{ov}^{h2}). \quad (2)$$

Through virtual attention heads, models can achieve complex functions by combining simple attention heads.

With *causal mask*, there is an interesting things when virtual attention heads do function like induction heads.

**Property 1.** *With causal mask and $j \geq i$, we have*

$$(A^{h2}A^{h1})_{j,i+1} = \sum_{k=1}^{N} A_{j,k}^{h2}A_{k,i+1}^{h1} = \sum_{k=i+1}^{j} A_{j,k}^{h2}A_{k,i+1}^{h1} \tag{3}$$

According to Property 3.2, from the perspective of virtual attention heads, it is difficult for the model to indirectly utilize $j$ tokens to focus on the $(i+1)^{th}$ token through $i^{th}$ token. In other words, in order for the $(i+1)^{th}$ token to be output by future tokens using the induced heads mechanism, it must first integrate the information of the $i^{th}$ token into its hidden states, even if the information of the $i^{th}$ token is useless for predicting the $(i+2)^{th}$ token. That's to say, **it imposes certain requirements on the dimensionality of hidden states**. We will further explain in the next section that the induction heads mechanism requires a certain width.

## 2.2. KV shifting attention

From a more general perspective, induction heads means obtaining information about some tokens by focusing on it surrounding tokens. In order to make it easier for the transformer to learn the mechanism of inductions heads, we can unbind the key and value of the $i^{th}$ token. When current token attention to the $i^{th}$ token's key, it can get the

$j^{th}$ token's value, $j \in \{i-1, i, i+1\}$. From a different perspective, current token can obtain the value of the $i^{th}$ token by focusing on the keys of the $\{i+1, i, i-1\}$ tokens.

However, if we directly use the combination of $j^{th}$ token's value ($j \in \{i-1, i, i+1\}$), it breaks causal mask, because $(i+1)^{th}$ token's value can't be computed when $i^{th}$ token try to do next token prediction. So we can only let $i^{th}$ token's key connect to $(i-1)^{th}$ and $i^{th}$ token's value. And if we do similar operation to value, which means let $i^{th}$ token's value connect to $(i-1)^{th}$ and $i^{th}$ token's key, we can find that we can pay attention to the key of $i^{th}$ token, and then get the value of its near token without breaking causal mask.

Therefore, we propose the following KV shifting attention. For simplicity, we only formalize the single head attention as follow:

$$Q, K, V = XW_Q, XW_K, XW_V \tag{4}$$

$$\hat{K}, \hat{V} = \alpha_1 K + \alpha_2 \text{Shift}(K), \beta_1 V + \beta_2 \text{Shift}(V) \quad (5)$$

$$\text{Output} = \text{Softmax}(Q\hat{K}^T \cdot M/\sigma)\hat{V}W_o, \tag{6}$$

where $X \in R^{N \times D}$ is hidden states, $W_Q, W_K, W_V, W_O \in R^{D \times D}$, $\alpha_1, \alpha_2, \beta_1, \beta_2 \in R$ are learnable parameters, $\sigma = \sqrt{D}$, $M \in R^{D \times D}$ is the causal mask, $\text{Shift}(\cdot)$ means discarding the last token and padding zero at the beginning. Compared to the original attention, 4 learnable parameters have been added. In the case of multi head attention, $\alpha_1, \alpha_2, \beta_1, \beta_2$ are learned per head, so the additional parameters is $4h$. The additional calculation caused by Eq 5 is $O(ND)$, which is much smaller than $O(ND^2 + N^2D)$ in Eq. 6.[1] We provide the training and inference code for PyTorch implementation in the Appendix M.

In order to ensure that the initialization of KV shifting does not affect the initial optimization state, we select $\alpha_1$ and $\beta_1$ from $\mathcal{U}(0,1)$ and let $\alpha_2 = 1 - \alpha_1$, $\beta_2 = 1 - \beta_1$.

## 3. Analysis

In this section, we will analyze the KV shifting attention. The first subsection will examine how KV shifting attention has a better ability to characterize induction heads compared to vanilla attention. The second subsection is to learn induction heads on the toy model and analyze their dynamic process. For more small experiments about the capability boundary of KV shifting attention, readers can refer to Appendix E. For an explanation of another perspective on KV shifting attention, please refer to Appendix 3.3.

---

[1] In current popular group queries attention (Ainslie et al., 2023) for large language models, the additional parameters is $4h_1$, where $h_1$ refers to the number of KV pairs, not the number of heads, and the additional calculation caused by Eq. (5) is $O(Nh_1d_1)$, where $d_1$ is the head dims.

## 3.1. Better representation for induction heads

The KV shifting attention reduced not only the depth but also width requirements of the transformers for forming induction heads mechanism. Improving the requirement for depth is very intuitive, while improving the requirement for width is the question we left in section 2.1. To strictly illustrate this point, we first modeling KV heads attention do induction heads. Then we use the theorem from previous work.

**Definition 3.1.** (Induction heads) We define the induction heads machine modeling by IH : $\bigcup_{L \in N^+} R^{L \times D} \mapsto R^D$ with Alibi RPE (Press et al.) as follows:

$$\text{IH}(x) = \sum_{s=2}^{L-1} \text{softmax} \left( x_L x_{s-1}^T / \sigma - m|L - s| \right)_i x_s. \quad (7)$$

Which has the ability to implement induction heads for any long context, when $T > 0$ and $m > 0$, which infinitely approach 0. In practice, transformers processed in a finite length, so only a very small $\sigma$ and $m$ can to be taken. [2]

Previous work (Wang et al., 2024) has shown that a two-layer transformer (Alibi RPE, without FFN) can be used to approximate **IH**, the detail is as follow:

**Theorem 3.2.** *(Modify from Wang et al. (2024)). There exists a constant $C > 0$ and a two-layer single-head transformer TF(without FFNs), with $D = 2d$, $W_K^{(1,1)} = W_Q^{(1,1)} = 0$, $p^{(2)} = m$, ($p^{(i)}$ means the Alibi bias in $i^{th}$ layers), and $\|W_K^{(2,1)}\|, \|W_Q^{(2,1)}\| \leq O(1, 1/\sigma)$, such that*

$$\sup_{L \in N^+} \|\text{IH} - TF\|_{L,\infty} \leq O(e^{-p^{(1)}}) \quad (8)$$

If we use KV shifting attention, we can get better estimation as follow:

**Theorem 3.3.** *There exists a **one**-layer single-head KV shifting attention KVSA, with $D = d$, such that*

$$\text{IH} = KVSA. \quad (9)$$

The proof in the case of using KV shifting attention is relatively simple if we are familiar with the induction heads. We include the proofs of Theorem 3.2 and Theorem 3.3 in Appendix A and Appendix B. From Theorem 3.2 and Theorem 3.3, we can find that KV shifting attention can represent or approximate induction heads with less depth and less width. In addition, since the copy operation in the first layer of the vanilla transformer introduces noise due to Alibi's bias, the final upper bound is bounded by a quantity

---

[2]We use infinite precision transformers in this article.

related to Alibi's bias. And the KV shifting attention, due to the absence of this noise, Eq. 9 takes an equal sign. [3]

For the theoretical upper limit of the transformer structure in induction heads or more generalized tasks, readers can refer to Sanford et al. (2024b).We believe that after replacing the standard Transformer with KV shifting attention, although there may not be a margin improvement in the theory bound, it is still possible to achieve a constant improvement. Next, we will leave the field of representation and enter the field of practice, which will provide experimental support for some of the assertion in this section.

## 3.2. Great bias when learning induction heads

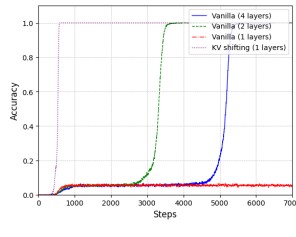 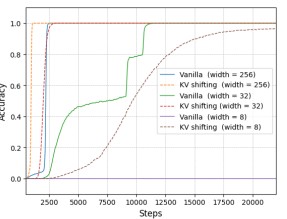

(a) Various depth          (b) Various width

Figure 1: On the left, as the training step size increases, the accuracy of induction varies among different models. In this setting, the only difference between Vanilla and KV shifting attention is the calculation of key and value. The total parameters of Vanilla and KV shifting attention with one layers is the same. And the parameters of Vanilla with 2 layers is twice. On the right is the induction accuracy with different hidden size. There are two layers in Vanilla model, and one layer in KV shifting attention, which means Vanilla model has two times parameters than KV shifting attention.

**Various depth** The architecture of the model adopts the same architecture as Llama (Touvron et al., 2023)[4] with approximately 20M no-embedding parameters. We used a huge vocabulary with 8000 tokens to randomly generate sentences and ensure that the sequences in them satisfy the condition that when the $j^{th}$ token is the same as the $i^{th}$ token, then the $(j + 1)^{th}$ token is the same as the $i^{th}$ token ($i < j$). We present the accuracy of $j^{th}$ next token prediction's accuracy in Figure 1a. The tasks is similar to MQAR (Arora et al., 2023), but the difference is that each token is a key for the next token and also a value for the previous token in our setting, while key value pairs are paired and disjoint in MQAR.

---

[3]Theorems 1 and 2 only provide constructive upper bounds for implementing induction heads. A more rigorous statement would be to prove that the lower bound of Theorem 1 is smaller than the upper bound of Theorem 2.

[4]These is a slightly different from the experiments in (Elhage et al., 2021), which use the attention-only structure, while we chose the Llama architecture for practice.

From Figure 1a, we can see that it is indeed difficult for a one-layer standard transformer to learn the ability of induction. And the 2-layer transformer and 1-layer KV shifting attention can perfectly learn the ability of induction. At the same time, KV shifting attention has a bias towards better learning induction, and its convergence speed is much faster than two-layer transformers. At the same time, we also found that increasing the model depth to layer 4 for Vallina does not make the model learn induction faster.

**Various width** Hidden size for the experiment in Figure 1a is set to 1024, which is a relatively relaxed setting. As we analyzed in sections 2.1 and 2.3. Existing attention may require additional width to perform well on induction heads, so as the dimensionality of hidden states decreases, it will become increasingly difficult for standard attention to learn induction heads. Therefore, we conducted pressure testing with hidden states=8. The results are shown in Figure 1b.

From Figure 1b, we find that the learning ability of standard attention is very poor, and even failed to cover up one of the answers mentioned in the previous text.

Although this is a toy task, people may think that the current model has a large dimension and can do the induction task well. But induction may also be done in some implicit way in language modeling. This limitation in width will result in the model considering a limited number of different implicit inductions in parallel, or introducing noise in superposition. Undoubtedly, this will have an negative impact on language modeling.

**Learning from induction data** Regarding how traditional two-layer transformers learn about induced heads, the analyzing it is a very complex. Previous work (Bietti et al., 2024; Wang et al., 2024; Chen et al., 2024) has provided analysis under simplified conditions. We also follow their work and provide an analysis of the dynamic process of KV shifting attention in learning induced heads. We use the following simplified conditions: removed the residual connection, MLP, normalization, position embedding and use tie embedding and the each component of the embedding of each token is from independently and identically distributed $\mathcal{N}(0, \frac{1}{d})$, $W_q, W_k, W_v, W_o = I$, and we assume the sentence's length is $T + 1$ and vocabulary size is $T$ ($T \geq 3$) and every token only appearance once except the last token appearance twice and calculate cross entropy loss only when predicting the next token on the last token. We analyzed the learning process of the four additional variables $\alpha_1, \alpha_2, \beta_1$, and $\beta_2$ introduced in KV shifting attention. We have:

**Theorem 3.4.** *Under the simplified condition we describe, and as d approx $\infty$, learning induction heads by KV shifting*

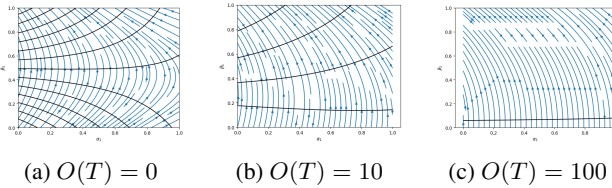

(a) $O(T) = 0$     (b) $O(T) = 10$     (c) $O(T) = 100$

Figure 2: Contour lines and gradient decent derection of $L$. We simplified $O(T)$ as a constant, and $\alpha_2 = 1 - \alpha_1$ and $\beta_2 = 1 - \beta_1$. Induction heads means $(\alpha_1, \beta_1) = (0, 1)$.

*attention is equivalent to:*

$$\min L =$$

$$- log \frac{e^{a_2\beta_1+\beta_2/S}}{e^{a_2\beta_1+\beta_2/S} + 2e^{\beta_1/S+\beta_2 a_2} + e^{2a_1\beta_1+a_2\beta_2} + O(T)},$$

(10)

*where $a_2 = e^{\alpha_2}/S$, $a_1 = e^{\alpha_1}/S$, $S = e^{\alpha_2} + 2e^{\alpha_1} + O(T)$, all $O(\cdot)$ here is greater than or equal to 0.*

The detail proof is in Appendix C. The dynamic process of optimization is interesting. We consider $O(T)$ as a constant and plot the contour lines and gradient directions of $-L$ for $\alpha_2 = 1 - \alpha_1$ and $\beta_2 = 1 - \beta_1$ in Figure 2.

From Figure 2, as $O(T)$ increases, the contour lines become sparse, which means the convergence speed slows down. In addition, when $O(T)$ is relatively small, the direction of GD may undergo a small non monotonic learning process. When $O(T)$ is relatively large, the direction of GD is relatively consistent. In practice, we often have many attention heads, $(\alpha_1, \beta_1)$ of some heads are closer to $(0, 1)$ during initialization, making it much easier for them to learn induction heads.

When $T = 2$, which is actually no longer applicable to Theorem 3.4, training data become limitation, such as [A] [B] [A] → [B]. The model only needs to predict the previous token to achieve low loss on the training data, but this is not an induction head. Learning such a head cannot achieve [A] [B] [C] [A] → [B].

In Bietti et al. (2024), they think global bigrams are learned first, then the induction head is formed by learning appropriate memories in a top-down fashion. But obviously, in KV shifting attention, induction heads become very easy to learn, and even with good initialization, the model has induction capability. But it is difficult to have appropriate initialization to obtain a certain level of bigrams capability without training.

There are some additional small-scale experiments on multi-hop, math and n-gram, which is in Appendix E.1, E.2 and E.3. Experiments have shown that performing shift operations on KV can significantly enhance abilities related to induction heads, such as multi-hop and math, but does not improve n-gram tasks that rely more on memory.

### 3.3. Meta learners and checker for in-context learning

In previous section, we present our primitive motivation, which is to reduce the depth and width requirements of induction heads through KV shifting attention. The formation of traditional induction heads mechanism requires the fusion of adjacent token information in the previous layer and the coupling of information into keys and values in the next layer, but fusion is easy and decoupling information is difficult, we think this may also be one of the reasons why Vanilla attention has a slower learning. In this section, we will provide another perspective on KV shifting attention.

Firstly, K shifting attention can be seen as a meta learner of in-context learning. For example, after undergoing K shifting, the KV pairs mentioned in the previous text are $(k_1, v_2), (k_2, v_3), ..., (k_{n-1}, v_n)$. Therefore, we can use the similarity between $k_n$ and historical $k_i$ to infer what $v_{n+1}$ is, as shown below:

$$v_{n+1} = \sum_{i=1}^{n-1} \frac{e^{k_n \cdot k_i / T}}{\sum_{i=1}^{n-1} e^{k_n \cdot k_i / T}} v_i. \qquad (11)$$

If we infer $v_{n+1}$, we can infer what the $(n+1)^{th}$ token is.

Then, we can find that V shifting attention can be seen as a checker of in-context learning. For example, after V shifting, we obtained such a pair of KV pairs, $(k_2, v_1), (k_3, v_2), ..., (k_n, v_{n-1})$, then if we guess the $(n+1)^{th}$ token is $x_{n+1}$, then we can use the guessed $k_{n+1}$ as query to do attention to infer $v_n$ (although we actually know what the real $v_i$ is) as follow:

$$v_n = \sum_{i=2}^{n} \frac{e^{k_{n+1} \cdot k_i / T}}{\sum_{i=1}^{n-1} e^{k_{n+1} \cdot k_i / T}} v_{i-1}. \qquad (12)$$

Then we can compare the guessed $v_n$ with the real $v_n$. If they are close, keep the guess for $x_{n+1}$, otherwise delete the guess for $x_{n+1}$ (keep or delete operation can be easily done in a Gate MLP layers and residual connection).

From this perspective, introducing KV shifting as an inductive bias into attention is much suitable for next token prediction based on the decode-only structure. Next, we will leave the toy models and validate the effectiveness of KV shifting attention on large-scale models.

## 4. Experiments

### 4.1. Setting

In this section, we evaluate KV shifting attention across two models trained from scratch: a 2.9B/19B parameters within an architecture similar to Llama2 (Touvron et al., 2023). The experiments trained from scratch are conducted on Nvidia H800-80G GPUs, while others are conducted on Nvidia A100-80G GPUs.

**Model Configuration** Since the baseline of the model trained from scratch is for production environments, not just for this paper, it uses Group query attention (GQA) (Ainslie et al., 2023) to reduce memory usage during inference and employs a larger vocabulary (48000) to cope with more multilingual environments. For this reason, our experiment on KV shfting attention, which was trained from scratch, was also based on the baseline. Instead of training the model from scratch, we used Llama's original vocabulary size of 36000 and employed standard Multi head attention (MHA). The detail configuration is as shown in Table 16. [5]

Table 1: Model Configuration.

| PARAMETERS | 1.5B | 2.9B | 6.7B | 13B | 19B |
|---|---|---|---|---|---|
| HIDDEN SIZE | 2,048 | 2,560 | 4,096 | 5,120 | 6,144 |
| LAYERS | 28 | 32 | 32 | 40 | 48 |
| HEAD NUMBER | 16 | 20 | 32 | 40 | 48 |
| KV NUMBER | 16 | 4 | 32 | 40 | 4 |
| FFN SIZE | 5,504 | 8,704 | 11,008 | 13,824 | 16,384 |
| MAX LENGTH | 2,048 | 4,096 | 2,048 | 2,048 | 12,288 |
| TOTAL TOKENS | 10B | 500B | 10B | 10B | 200B |
| VOCAB SIZE | 36,000 | 48,000 | 36,000 | 36,000 | 48,000 |

**Datasets** Due to some commercial reasons and data limitations, we used non-public private data for pre training, which means that the models are trained on our own dataset. Our data collection and filtering methods are similar to FineWeb-edu (Penedo et al., 2024). When computing resources are available, we will use open-source data, such as RedPajama-1T(Weber et al.) to train two models for comparison, one is the baseline and the other is the KV shifting attention.

**Hyperparameters** We used a constant learning rate with a linear warmup of 1000 steps. The learning rate for 1.4B / 3B / 7B / 13B / 19B model is 2e-4 / 8e-4 / 2e-4 / 2e-4 / 2e-4, the batch size is 1M/16M/1M/2M/3M[6]. For optimization, we apply the AdamW optimizer with $\beta_1 = 0.9$ and $\beta = 0.95$, and weight decay = 0.1.

**Evaluation** To validate the performance of different models, we used some benchmarks for the 3B and 19B models that were trained more tokens, including Lambada(Paperno et al., 2016), Winogrande(Sakaguchi et al., 2021), Hellaswag(Zellers et al., 2019), ARC(Clark et al., 2018), CMMLU(Li et al., 2023a), MMLU(Hendrycks et al.), Math(Hendrycks et al., 2021). While for the models that were trained less token, we could only look at their loss curve. Due to computation limitations, almost experiments are only run once except there is an additional notion.

---

[5]The Vanilla-2.9B and KV shifting-2.9B has been open-sourced.

[6]1M here means 1,048,576 = 1,024 × 1,024, while elsewhere in the paper it refers to 1,000,000

Table 2: Main results. We trained four models on 2.9B and 19B parameters respectively, with the 2.9B model having a total training token count of 500B and the 19B model having a total training token count of 200B. ARC-E is short for ARC-easy, and ARC-C is short for ARC-Challenge.

| MODEL | TOKENS | LAMBADA | WINOGRANDE | HELLASWAG | ARC-E | ARC-C | CMMLU | MMLU | MATH | AVERAGE |
|---|---|---|---|---|---|---|---|---|---|---|
| VANILLA - 2.9B | 340B | 52.92 | 52.09 | 42.70 | 27.45 | 25.97 | 28.51 | 29.43 | 0.80 | 32.48 |
| | 420B | 52.80 | 54.85 | 43.68 | 28.96 | 26.02 | 34.77 | 30.34 | 1.20 | 34.08 |
| | 500B | 51.66 | 54.06 | 44.49 | 36.20 | 27.90 | 38.22 | 37.26 | 1.80 | 36.45 |
| KV SHITING - 2.9B | 340B | **55.44** | 53.91 | 42.87 | 36.74 | 30.04 | 34.51 | 36.20 | 2.00 | 36.46 |
| | 420B | 51.91 | 54.78 | 43.83 | 36.66 | **31.91** | 37.24 | 34.30 | 1.80 | 36.55 |
| | 500B | 54.51 | **55.33** | **44.52** | **39.02** | 30.89 | **40.78** | **40.88** | 2.60 | **38.57** |
| VANILLA - 19B | 160B | 59.93 | 48.22 | 48.25 | 30.34 | 24.56 | 39.12 | 39.22 | 1.80 | 36.43 |
| | 180B | 58.80 | 48.07 | 47.78 | 31.28 | 25.99 | 40.80 | 39.34 | 2.60 | 36.83 |
| | 200B | 60.88 | **49.01** | 47.36 | 33.25 | 25.78 | 42.92 | 42.68 | 2.60 | 38.06 |
| KV SHIFTING - 19B | 160B | 61.93 | 48.46 | **48.28** | 31.25 | 25.06 | 42.10 | 42.87 | 2.00 | 37.74 |
| | 180B | 60.20 | 47.67 | 48.16 | 32.45 | 26.55 | **43.38** | 40.49 | 3.00 | 37.74 |
| | 200B | **62.35** | 48.38 | 48.42 | **33.28** | **29.32** | 42.40 | **43.29** | **3.20** | **38.83** |

## 4.2. Main Result

We pre trained language models at two scales with 2.9B and 19B parameters, and the experimental results are shown in Table 2. The results indicate that KV Shifting attention achieved better performance than baseline at various scales and training token numbers. In addition, we also plotted the training loss of them, as shown in Figure 3.

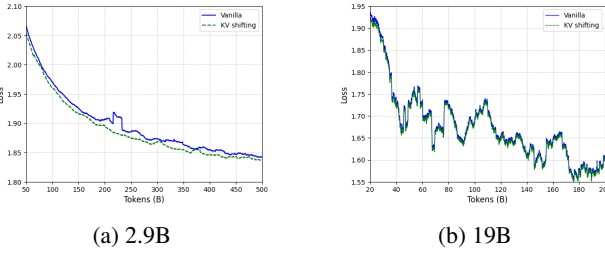

(a) 2.9B                                    (b) 19B

Figure 3: Training loss curve. We train 2.9B model with 500B tokens, and 19B models with 200B tokens.

In Table 2, we can find that KV shifting attent KV Shifting introduces a bias that is more suitable for language modeling, accelerating the convergence of the model. And it seems that both 2.9B models are converge on the benchmark Lambda, and KV shifting attention can achieve better results. Here we slightly argue that KV shifting attention can achieve better performance than the vanilla model when they both converge, although it may take several TB data for a 2.9B model to converge.

## 4.3. Robust Experiment

To verify the robustness of the model, we used different random seeds for 1.5B model, specifically, we set random number seeds for model initialization and data sampling, and conducted five experiments. The Vanilla and KV shifting attention in each experiment using the same re initialization and data. As shown in Figure 4a, although the training loss is quite shaky, it can be seen that under different random seeds, KV shifting attention is always better than vanilla.

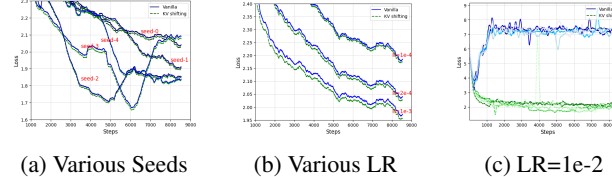

(a) Various Seeds          (b) Various LR          (c) LR=1e-2

Figure 4: Training loss of 1.5B parameters model among random seeds and learning rate (LR).

In addition, we also conducted experiments on different learning rates, including 1e-4, 2e-4, 1e-3, and 1e-2, as shown in Figure 4b. It can be observed that KV shifting attention achieves better results than Vanilla at different learning rates. And when the learning rate is set to 1e-2, Vanilla has already diverged, while the loss of KV shifting attention has not yet diverged as shown in Figure 4c[7]. This suggests that the optimization space for KV shifting attention may be flatter. Shifting KV provides a smooth key and value during model initialization, which may make the optimization process smoother. This also partially explains why in Figure 3, the 2.9B KV shifting attention leads more in terms of loss, because the 2.9B model is trained with a large learning rate[8].

## 4.4. Scaling Experiment

Firstly, we plotted the training loss curves at the 1.5B, 6.7B, and 13B parameters as shown in Figure 5. We found that under different settings, KV shifting attention achieved better results compared to vanilla model.

Afterwards, we also followed Kaplan et al. (2020) and used WebText (Radford et al.) as the validation set to draw a scaling law for vanilla and KV shifting attention. As shown

---

[7]To confirm this, we attempted 5 random seeds under the condition of LR=1e-2, and the all results of each experiment are Vanilla divergence and KV shifting convergence.

[8]We adopt a large learning rate in practice, because Lobacheva et al. suggests large learning rates improve generalization

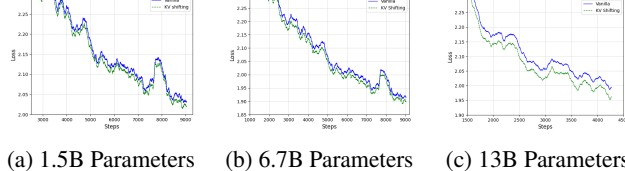

(a) 1.5B Parameters    (b) 6.7B Parameters    (c) 13B Parameters

Figure 5: Training loss comparison between different size. All models are trained on 10B tokens. The batch size for 1.5B and 6.7B model is 0.5M, for 13B is 1M, so the total steps of 13B model is half of others.

in Figure 6a[9], KV shifting attention also has excellent scaling properties and outperforms baseline at every parameter scale. And we continued to train the 1.5B model to 30B tokens, which we calculate the validation set loss every 1000 steps, but the difference in validation loss still does not decrease, as shown in Figure 6b.

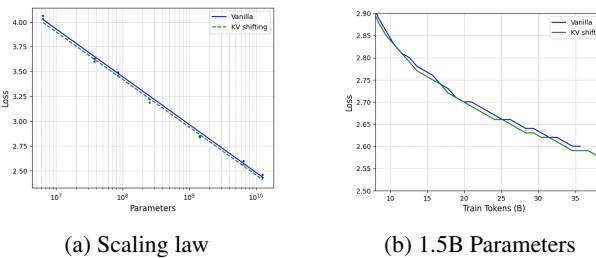

(a) Scaling law      (b) 1.5B Parameters

Figure 6: Validation loss across different size and training tokens. For scaling law, while others in this paper means the total parameters. models are trained on 10B tokens and calculate the final checkpoint's validation loss.

### 4.5. Further experiments

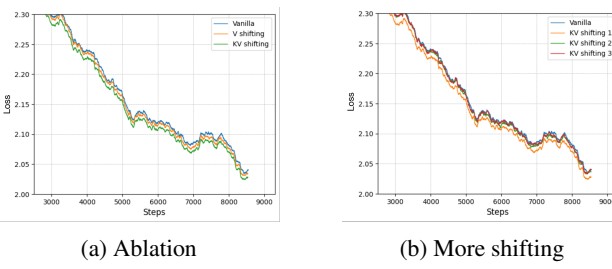

(a) Ablation      (b) More shifting

Figure 7: Further experiments are conducted on a 1.5B model, where we trained 10B tokens.

To further validate the effectiveness of the proposed KV shifting attention, we conducted the following ablation experiments. Firstly, in the experiment of ablating the shifting of $K$ and $V$, as shown in Figure 7a, we find that the shifting of Key and Value both plays an important role. It can be inferred that obtaining the value of the $(i-1)^{th}$ token or the value of the $(i+1)^{th}$ token by focusing on the key of the

$i^{th}$ token is important in language modeling. If K shifting or V shifting is not used, the model needs to use two layers of attention to implement the function. And we provide another way to explain the role of K shifting and V shifting in Appendix 3.3.

In addition, we also performed longer shifts on K and V, extending the shift between $i-1$ and $i$ to $[i-2, i]$ and $[i-3, i]^{10}$. The results are shown in Figure 7b. The results have shown that using a longer shift window does not improve performance, although increases computational complexity. From the perspective of induction heads, using our current shifting window size is sufficient. If someone want to expand to longer windows, it may need to carefully design it. In this paper, we will no longer attempt more refined designs, as current lightweight designs are sufficient for better learning of induction heads and improving language modeling. And we also conducted a shift operation on all QKV in Appendix G and H.

## 5. Discussion

Overall, based on previous research on the induction heads mechanism in transformer model, we have designed an attention mechanism that is more suitable for learning induction heads. This can not only reduce the demand for width, but also reduce the demand for layers to learn induction heads. At the same time, in large-scale language model pre training, the use of KV shifting attention can achieve better results than baseline in many experimental settings, which implicitly demonstrates the importance of learning induction heads for language modeling.

Another noteworthy fact is that the previous research (Elhage et al., 2021; Akyürek et al., 2024) think transformer to outperform LSTMs at in-context learning on text, with induction heads as a major explanation. Our experiments indicate **that existing transformer mechanisms still have not fully unleashed the potential of learning induction heads.** By making slight modifications to the original architecture, the model can learn induction heads much faster than before. When the model can learn induction heads in a simple way, it may becomes easier to achieve length generalization (Zhou et al., 2024). And it may be interesting to delve into the underlying mechanisms of transformer, identify important features, and modify the transformer to make it easier to implement important functions. For example, it is very challenging to learn parity check (Wies et al., 2023).

Due to our lightweight modifications, the KV shifting attention can **easily be compatible with existing training and inference frameworks**. We believe that there is not only

---

[9]In this figure, the parameters is the no-embedding parameters, followed by Kaplan et al. (2020), while others in this paper mean total parameters.

[10]KV shifting 1 is the KV shifting we used in our paper. For KV shifting 2, we randomly initialize $\alpha_1, \alpha_2$ from $U(0,1)$, and then use $1 - \alpha_1 - \alpha_2$ as $\alpha_3$. It is similar for $\beta$ and KV shifting 3.

one way for models to better learn induction heads. But some complex operations may be difficult to adapt to under existing training or inference acceleration frameworks.

From the perspective of operation, the fusion of adjacent information in neural network has a long history, such as Wu et al. (2018) in vision, Zhang et al. (2021) in video, Li et al. (2023b) in speech, and Cobbe et al. (2021) in text. Many of them fuse or shift all information rather than KV. However, existing LLMs may actually easily integrate adjacent token information without introducing additional inductive biases due to position embedding and causal mask. Some works (Chou et al., 2024) also believe that such operations will enhance local interactions. However, as shown in the Appendix E.3, we conduct a 3-gram experiment and the addition of KV shifting does not improve the learning ability of 3-grams, which also implies that **the key to the effectiveness of KV shifting attention is not to promote the fusion of local information which is usually considered an important role of convolution, but to achieve better induction heads by decoupling key-value pairs in time sequence**, thereby enhancing in-contextual learning. The fusion of information is easy, but the decoupling of information may be difficult. We believe our work can enhance the understanding of the use of shift operations in attention.

The shifting operation can be seen as a short convolution with kernel size 2. We can see that there are some interesting connections with another research line in languages modeling, which is the extensive use of short convolution structures in non transformer architecture , such as RWKV (Peng et al., 2023), H3 (Fu et al., 2022), Hyena (Poli et al., 2023), Mamba series (Gu & Dao, 2023; Dao & Gu), xLSTM (Beck et al., 2024), and DeltaNet (Yang et al., 2024b). The short convolutions can be a component in improving performance, especially in enhancing their in-context learning ability (Yang et al., 2024b). We believe that our work can provide some insight, such as performing short convolutions on queries may have a small contribution to in-context learning, one possible alternative is to set the kernel size of the query's short convolution to key's -1 to simulate the learning process of n-gram induction heads. However, from the perspective of induction heads, it is difficult to explain the role of short convolutions, because generalized state space models (GSSMs) cannot arbitrarily retrieve from context (Jelassi et al., 2024).[11] We can try using a hybrid model if we really want to perform it, and whether it is necessary to apply short convolutions in the non transformer part of the hybrid structure in the presence of attention layers that are proficient in retrieval may be an interesting question.

Besides, using KV shifting may be much easier to locate the induction heads, which can be helpful for mechanistic

---

[11]For fixed state space, it can perfectly achieve ...[S][B]... [S] -> [B], where [S] is a special token rather a arbitrary token.

interpretability. If we start from making the model more interpretable, we can also impose some constraints (Friedman et al., 2024), but this may not improve the performance.

## 6. Related works

**Induction Heads**   Induction heads, introduced by Olsson et al. (2022); Elhage et al. (2021), are specialized attention mechanisms that identify patterns in sequences, enabling large language models to predict subsequent tokens based on previous patterns. There are numerous studies (Bansal et al., 2023; Conmy et al., 2023; Wang et al.; Ren et al., 2024; Todd et al.) starting from induction heads to investigate the interpretability of large models, in order to enhance their transparency. Another part of works (Bietti et al., 2024; Wang et al., 2024; Sanford et al., 2024b; Chen et al., 2024) are to study how the model learns induction heads from a theoretical perspective. Previous studies motivated us greatly, and the focus of our work is to modify the attention to make the model better learn induction heads.

**Model Structure and Language Modeling**   Over the past many years, people have attempted various structures to enhance model's language modeling abilities. From early RNN (Mikolov et al., 2010) and LSTM (Sundermeyer et al., 2012) to the dominant transformer (Vaswani, 2017) today. Transformers have quadratic complexity, and many efforts have been made to improve them, such as RWKV (Peng et al., 2023), Mamba (Gu & Dao, 2023), RetNet (Sun et al., 2023). However, transformers still have many excellent properties that cannot be replaced temporarily, especially their ability to retrieve information (Jelassi et al., 2024; Alman & Yu, 2024). Currently, popular language models (Achiam et al., 2023; Dubey et al., 2024; Yang et al., 2024a) still use Transformers as architecture. Recently, there have been also some efforts to modify transformers to enhance modeling capabilities, such as reducing the noise of attention (Ye et al., 2024a) or reducing attention to unneeded elements (Leviathan et al., 2024). Our work is to slightly modify attention to enhance its ability to learn induction heads, which potentially improves language modeling.

## 7. Conclusion

In this work, we analyzed that induction heads have certain requirements for both the width and depth of the transformer, which motivated us to implement the induction mechanism more efficiently. For stronger expressiveness and faster convergence for the learning induction heads, we propose the KV shifting attention, enhancing the language modeling ability of the decode-only structure transformers. We conducted extensive experiments to verified the effectiveness of KV shifting attention, including pre training of 2.9B and 19B parameter models. We hope this work can provide some inspiration for achieving more powerful language modeling

or enhancing understanding of transformers.

## Impact statement

This paper presents work whose goal is to advance the field of langugage modeling. There are many potential societal consequences of our work, none which we feel must be specifically highlighted here.

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

## A. Proof of Theorem 3.2

Proof. We consider two-layer single-head transformer without FFN, where the first layer has the residual block, while the second layer does not have the residual block.

We first embed each token into $R^D$ as $\begin{pmatrix} x_s \\ 0 \end{pmatrix}$ and take $W_V^{(1)} = \begin{pmatrix} 0 & 0 \\ I_{d \times d} & 0 \end{pmatrix}$, then the s-th output token of the first layer is

$$\begin{pmatrix} x_s \\ y_s \end{pmatrix} = \begin{pmatrix} x_s \\ \sum_{\tau=1}^{s-1} \text{softmax}\left(-p^{(1)}(s-1-\tau)\right) x_\tau \end{pmatrix}.$$

Then for the second layer, we choose $p^{(2)} = m$,

$$W_Q^{(2,1)} = \begin{pmatrix} 0 & 0 \\ I_{d \times d} & 0 \end{pmatrix}, W_K^{(2,1)} = \begin{pmatrix} 0 & 0 \\ 0 & W^\star \end{pmatrix}, W_V^{(2,1)} = \begin{pmatrix} I_{d \times d} & 0 \\ 0 & 0 \end{pmatrix} \in \mathbb{R}^{D \times D},$$

and the projection $W_O^{(2)} = (I_{d \times d}, 0_{d \times d}) \in \mathbb{R}^{d \times D}$. Then the last output token of the second layer is

$$\sum_{s=2}^{L-1} \text{softmax}(x_L^\top y_s / \sigma - m|L-s|)x_s$$

By Lemma D.1, for any $L \in N^+$

$$\begin{aligned}
&\|\mathbf{IH}_2 - \mathbf{TF}\|_{L,\infty} \\
&= \sup_{X_L} \|\mathbf{IH}(X_L) - \mathbf{TF}_{-1}(X_L)\|_\infty \\
&= \left\| \sum_{s=2}^{L-1} \text{softmax}(x_L^\top y_s / \sigma - m|L-s|)x_s \right. \\
&\quad \left. - \sum_{s=2}^{L-1} \text{softmax}(x_L^\top x_{s-1}/\sigma - m|L-s|)x_s \right\|_\infty \\
&\leq \sum_{s=2}^{L-1} |\text{softmax}(x_L^\top y_s/\sigma - m|L-s|) \\
&\quad - \text{softmax}(x_L^\top x_{s-1}/\sigma - m|L-s|)| \\
&\leq 2\sup_s |x_L^\top y_s/\sigma - x_L^\top x_{s-1}/\sigma| \\
&\leq 2\|x_L^\top/\sigma\|_1 \sup_s \|y_s - x_{s-1}\|_\infty \\
&\leq 2 \sum_{i,j} |I/\sigma| \sup_s \left\| \left( \sum_{\tau=1}^{s-1} \text{softmax}\left(-p^{(1)}(s-1-\tau)\right) x_\tau \right) - x_{s-1} \right\|_\infty \\
&\leq 2\|I/\sigma\|_{1,1} \sup_s \left\| \left( \text{softmax}\left(-p^{(1)}((s-1-\tau))\right) \right)_{\tau=1}^{s-1} - e_{s-1} \right\|_1 \\
&= 4\|I/\sigma\|_{1,1} \sup_s \left| \frac{1}{\sum_{\tau=0}^{s-2} e^{-p^{(1)}\tau}} - 1 \right| \\
&< 4\|I/\sigma\|_{1,1} \frac{e^{-p^{(1)}}}{1 - e^{-p^{(1)}}} \leq O\left(e^{-p^{(1)}}\right).
\end{aligned}$$

## B. Proof of Theorem 3.3

If we set $\alpha_1 = 0$, $\alpha_2 = 1$, $\beta_0 = 1/\sigma$, $\beta_2 = 0$, $p = m$ (the bias of Alibi position embedding), $W_q = W_k = W_o = W_v = I \in R^{d \times d}$, we can get the proof.

## C. Proof of Theorem 3.4

We consider embeddings $u_k \in R^d$ with i.i.d. Gaussian $\mathcal{N}(0, \frac{1}{d})$ entries.

We recall a few facts:

- (Norm) We have $u_i^\top u_i = 1 + O(1/\sqrt{d})$. Because $u_i^\top u_i$ is a scaled chi-squared distribution, with mean = 1, and variance $2/d$.

- (Near-orthogonality) For $i \neq j$, we have $u_i^\top u_j = O(1/\sqrt{d})$. To see this, denoting $u_i = d^{-1/2}(\tilde{u}_{ik})_k$, where $\tilde{u}_{ik}$ are the normalized entries of $u_i$, note that we have

$$\sqrt{d}u_i^\top u_j = \frac{1}{\sqrt{d}} \sum_{k=1}^{d} \tilde{u}_{ik}\tilde{u}_{jk} \to \mathcal{N}(0, 1),$$

by the central limit theorem, since for each $k$, the quantities $\tilde{u}_{ik}\tilde{u}_{jk}$ are zero-mean, unit-variance, i.i.d. random.

Although we can describe it more accurately, in (10) we actually only used when $d$ tends towards positive infinity, $u_i^\top u_i = 1$ and $u_i^\top u_j = 0$.

Assume the sequence is $x_1, x_2, ..., x_i, x_{i+1}, ..., x_T, x_{T+1}$, where $x_{T+1} = x_i$, and $x_1, x_2, ..., x_T$ are different.[12] Now let's start our proof:

**Calculate attention score** The last token attend to the $k^{th}$ token, the attention score $\hat{a}_k$ before softmax can be calculate as:

If $k = i$, $\hat{a}_k = x_i^\top(\alpha_1 x_i + \alpha_2 x_{i-1}) = \alpha_1 + O(\frac{\alpha_1 + \alpha_2}{\sqrt{d}})$;

If $k = i + 1$, $\hat{a}_k = x_i^\top(\alpha_1 x_{i+1} + \alpha_2 x_i) = \alpha_2 + O(\frac{\alpha_1 + \alpha_2}{\sqrt{d}})$;

If $k = T + 1$, $\hat{a}_k = x_i^\top(\alpha_1 x_i + \alpha_2 x_n) = \alpha_1 + O(\frac{\alpha_1 + \alpha_2}{\sqrt{d}})$;

Other else, $\hat{a}_k = x_i^\top(\alpha_1 x_k + \alpha_2 x_{k-1}) = O(\frac{\alpha_1 + \alpha_2}{\sqrt{d}})$.

So, the attention score $a_k$ after softmax can be calculate as: If $k = i$, $a_k = e^{\alpha_1}O(e^{\frac{\alpha_1 + \alpha_2}{\sqrt{d}}})/S$;

If $k = i + 1$, $a_k = e^{\alpha_2}O(e^{\frac{\alpha_1 + \alpha_2}{\sqrt{d}}})/S$;

If $k = T + 1$, $a_k = e^{\alpha_1}O(e^{\frac{\alpha_1 + \alpha_2}{\sqrt{d}}})/S$;

Other else, $a_k = O(e^{\frac{\alpha_1 + \alpha_2}{\sqrt{d}}})/S$, where $S = 2e^{\alpha_1}O(e^{\frac{\alpha_1 + \alpha_2}{\sqrt{d}}}) + e^{\alpha_2}O(e^{\frac{\alpha_1 + \alpha_2}{\sqrt{d}}}) + O(Te^{\frac{\alpha_1 + \alpha_2}{\sqrt{d}}})$.

**Calculate logits** Now we calculate the logits of predict $k^{th}$ token, donate as $l'_k$:

If $k' = i$, $l_{k'} = \sum_{k=1}^{T+1} x_i^\top(\beta_1 x_k + \beta_2 x_{k-1})a_k = (\beta_1 + O(\frac{\beta_1 + \beta_2}{\sqrt{d}}))a_i + (\beta_2 + O(\frac{\beta_1 + \beta_2}{\sqrt{d}}))a_{i+1} + (\beta_1 + O(\frac{\beta_1 + \beta_2}{\sqrt{d}}))a_{T+1} + O(T)O(\frac{\beta_1 + \beta_2}{\sqrt{d}})O(e^{\frac{\alpha_1 + \alpha_2}{\sqrt{d}}})/S = (2\beta_1 e^{\alpha_1} + 2e^{\alpha_1}O(\frac{\beta_1 + \beta_2}{\sqrt{d}}) + \beta_2 e^{\alpha_2} + e^{\alpha_2}O(\frac{\beta_1 + \beta_2}{\sqrt{d}}) + O(\frac{(\beta_1 + \beta_2)T}{\sqrt{d}}))O(e^{\frac{\alpha_1 + \alpha_2}{\sqrt{d}}})/S$. (Expand according to k=i, i+1, T+1, other else.)

If $k' = i + 1$, $l_{k'} = \sum_{k=1}^{T+1} x_{i+1}^\top(\beta_1 x_k + \beta_2 x_{k-1})a_k = O(\frac{\beta_1 + \beta_2}{\sqrt{d}})a_i + (\beta_1 + O(\frac{\beta_1 + \beta_2}{\sqrt{d}}))a_{i+1} + (\beta_2 + O(\frac{\beta_1 + \beta_2}{\sqrt{d}}))a_{i+2} + O(\frac{\beta_1 + \beta_2}{\sqrt{d}})a_{T+1} + O(T)O(\frac{\beta_1 + \beta_2}{\sqrt{d}})O(e^{\frac{\alpha_1 + \alpha_2}{\sqrt{d}}})/S = (2e^{\alpha_1}O(\frac{\beta_1 + \beta_2}{\sqrt{d}}) + \beta_1 e^{\alpha_2} + e^{\alpha_2}O(\frac{\beta_1 + \beta_2}{\sqrt{d}}) + \beta_2 + O(\frac{(\beta_1 + \beta_2)T}{\sqrt{d}}))O(e^{\frac{\alpha_1 + \alpha_2}{\sqrt{d}}})/S$ (Expand according to k=i, i+1, i+2, T+1, other else.)

If $k' = T$, $l_{k'} = \sum_{k=1}^{T+1} x_T^\top(\beta_1 x_k + \beta_2 x_{k-1})a_k = O(\frac{\beta_1 + \beta_2}{\sqrt{d}})a_i + O(\frac{\beta_1 + \beta_2}{\sqrt{d}})a_{i+1} + (\beta_1 + O(\frac{\beta_1 + \beta_2}{\sqrt{d}}))a_T + (\beta_2 + O(\frac{\beta_1 + \beta_2}{\sqrt{d}}))a_{T+1} + O(T)O(\frac{\beta_1 + \beta_2}{\sqrt{d}})O(e^{\frac{\alpha_1 + \alpha_2}{\sqrt{d}}})/S = (e^{\alpha_1}O(\frac{\beta_1 + \beta_2}{\sqrt{d}}) + e^{\alpha_2}O(\frac{\beta_1 + \beta_2}{\sqrt{d}}) + \beta_1 + \beta_2 e^{\alpha_1} + O(\frac{(\beta_1 + \beta_2)T}{\sqrt{d}}))O(e^{\frac{\alpha_1 + \alpha_2}{\sqrt{d}}})/S$ (Expand according to k=i, i+1, T, T+1, other else.)

If $k' = i - 1$, $l_{k'} = \sum_{k=1}^{T+1} x_{i-1}^\top(\beta_1 x_k + \beta_2 x_{k-1})a_k = (\beta_1 + O(\frac{\beta_1 + \beta_2}{\sqrt{d}}))a_{i-1} + (\beta_2 + O(\frac{\beta_1 + \beta_2}{\sqrt{d}}))a_i +$

---

[12] Here, we assume $i > 1$ and $i < T$, if not, there will be a slight difference in proof.

$O(\frac{\beta_1+\beta_2}{\sqrt{d}})a_{i+1} + O(\frac{\beta_1+\beta_2}{\sqrt{d}})a_{T+1} + O(T)O(\frac{\beta_1+\beta_2}{\sqrt{d}})O(e^{\frac{\alpha_1+\alpha_2}{\sqrt{d}}}))/S = (\beta_1 + \beta_2 e^{\alpha_1} + e^{\alpha_2}O(\frac{\beta_1+\beta_2}{\sqrt{d}}) + 2e^{\alpha_1}O(\frac{\beta_1+\beta_2}{\sqrt{d}}) + O(\frac{(\beta_1+\beta_2)T}{\sqrt{d}}))O(e^{\frac{\alpha_1+\alpha_2}{\sqrt{d}}}))/S$ (Expand according to k=i-1, i, i+1, T+1, other else.)

Other else, $l_{k'} = \sum_{k=1}^{T+1} x_{k'}^\top(\beta_1 x_k + \beta_2 x_{k-1})a_k = O(\frac{\beta_1+\beta_2}{\sqrt{d}})a_i + O(\frac{\beta_1+\beta_2}{\sqrt{d}})a_{i+1} + (\beta_1 + O(\frac{\beta_1+\beta_2}{\sqrt{d}}))a_{k'} + (\beta_2 + O(\frac{\beta_1+\beta_2}{\sqrt{d}}))a_{k'+1} + O(T)O(\frac{\beta_1+\beta_2}{\sqrt{d}})O(e^{\frac{\alpha_1+\alpha_2}{\sqrt{d}}}))/S = (e^{\alpha_1}O(\frac{\beta_1+\beta_2}{\sqrt{d}}) + e^{\alpha_2}O(\frac{\beta_1+\beta_2}{\sqrt{d}}) + \beta_1 + \beta_2 + O(\frac{(\beta_1+\beta_2)T}{\sqrt{d}}))O(e^{\frac{\alpha_1+\alpha_2}{\sqrt{d}}}))/S$ (Expand according to k=i, i+1, k', k'+1, other else.)

As $d$ tending towards positive infinity, we summay the logits of $i^{th}$ token in Table 3:

| predict tokens | logit |
|---|---|
| i - 1 | $(\beta_1 + \beta_2 e^{\alpha_2})/S$ |
| i | $(2\beta_1 e^{\alpha_1} + \beta_2 e^{\alpha_2})/S$ |
| i+1 | $(\beta_1 e^{\alpha_2} + \beta_2)/S$ |
| T | $(\beta_1 + \beta_2 e^{\alpha_2})/S$ |
| other else | $(\beta_1 + \beta_2)/S$ |

Table 3: Logits summary, when $d$ tend to $\infty$, where $S = 2e^{\alpha_1} + e^{\alpha_2} + O(T)$

**Calculate loss**   We denote $e^{\alpha_1}/S$ as $a_1$, $e^{\alpha_2}/S$ as $a_2$, than we can get:

$$Loss = -log(\frac{e^{l_{i+1}}}{\sum_{i=1}^{T} e^{l_i}}) \tag{13}$$

$$= -log(\frac{e^{a_2\beta_1+\beta_2/S}}{e^{a_2\beta_1+\beta_2/S} + 2e^{\beta_1/S+\beta_2 a_2} + e^{2a_1\beta_1+a_2\beta_2} + O(T)}) \tag{14}$$

## D. Limitation

Due to limitations in computing resources and other limitations, many experiments cannot be repeated many times, and it is not convenient for us to use open-source datasets or open source our datasets. But our robustness experiments have convinced us that KV shifting attention can achieve better results in many experimental settings, and we have also open-source our pertained model.[13] In theory, we provide a way for KV shifting attention to learn induction heads under relaxed conditions. However, it remains challenging on how to learn induction heads under multi-layer transformer.

Meanwhile, induction heads can actually be implemented in another complex way. For example, the model can focus on similar tokens, obtain the hidden location information of the $i^{th}$ position, convert this information into the information of the $(i+1)^{th}$ position through MLP, and then use it as query information in the next layer, so as to focus on the information of the $(i+1)^{th}$ token. It is currently unclear how this kind of induction heads affects the training dynamics of the model. However, if the key of attention is allowed to shift in the next layer, there is no need for MLP to convert $i^{th}$ position information into the $(i+1)^{th}$, which maybe also be helpful for the learning of induction heads.

## E. Addition experiments to exploration the capability boundary

### E.1. Hop k

In this section, we evaluate the a multi-layered form of induction heads, namely multi-hop. We followed the code of Sanford et al. (2024b) and only changed the attention of vanilla to KV shifting. The experimental results are shown in Figure 8. Obviously, KV shifting has a very good bias for learning multi-hop.

Taking the pink line (L=5) as an example, the performance of the vanilla model will significantly degrade when the hop number exceeds 8. However, the KV shifting attention still has a small error when the hop count reaches 16. This powerful ability to perform implicit reasoning implies that KV shifting attention may achieve better results in mathematical or reasoning abilities. Therefore, we present the experimental results of mathematical ability in the next section.

---

[13]There is an open source implements of KV shifting attention. https://github.com/erogol/BlaGPT. If you want to verify the effectiveness of KV shifting attention, using their code and spending 8 NVIDIA-A100 plus 2 hours of training time is a good choice.

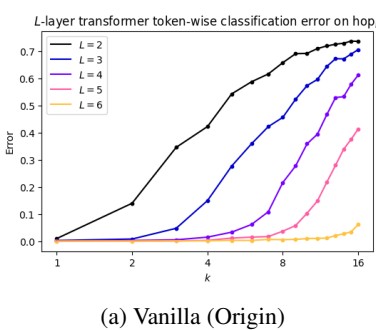
(a) Vanilla (Origin)

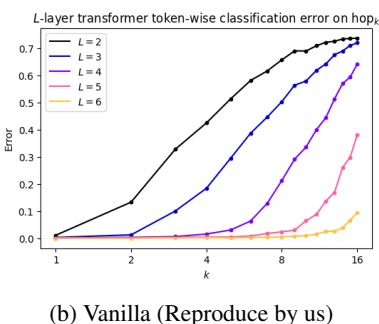
(b) Vanilla (Reproduce by us)

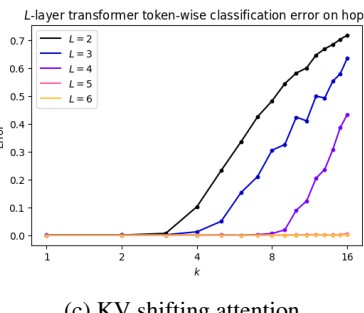
(c) KV shifting attention

Figure 8: Comparison of Error Rates under Hop k Tasks. The smaller the error, the better the performance.

### E.2. Grade-School Math

As a more direct manifestation of induction ability, learning math problems is a natural experiment. At the same time, in order to eliminate the influence of complex syntax, test data leakage, etc. We follow Ye et al. (2024b) and conduct experiments on the iGSM dataset. Due to the code of Ye et al. (2024b) haven't been released yet, we start from code of **https://github.com/kaminocode/iGSM**. We use the similar hyper-parameters, except the context length which we set as 1024 for all experiments and the learning rate which we set as 2e-4. [14]

In addition, as we are using an open-source code implementation. For simplicity, there is no final summary of the answer. So the evaluation of accuracy is based on the fact that as long as some step calculates the question asked and answers the correct answer, it is considered to be answered correctly, even if additional calculation steps are performed after answering the correct answer.

Table 4: Experiments on iGSM. **Tr X - Te Y** means train with the numbers of operation no greater than X and test with the numbers of operation as Y.

| MODEL | TR12-TE15 | TR21-TE24 |
|---|---|---|
| VANILLA | 0.8154 | 0.8711 |
| KV SHIFTING | 0.8909 | 0.9062 |

The result in Table 4 shown the enormous potential of KV shifting attention. We expect KV shifting attention to enhance the reasoning ability of the model by improving its ability in basic induction heads.

We have included this section in the appendix because our experiment was not as thorough as Ye et al. (2024b). We only tested the accuracy without delving deeper into the analysis, such as the reasons for mistake like Ye et al. (2024b). Besides, conducting more in-depth experiments is beyond the scope of this article.

### E.3. Can KV shifting attention learn n-gram better?

In addition to induction ability, an important part of language modeling is n-gram In this section, we test the model's ability to learn n-grams. We randomly generated approximately 200 pairs of $x_1$ and $x_2$, and then we randomly generated $x_3$ for each pair. The model needs to accurately predict $x_3$ when seeing $x_1$ and $x_2$. The results are shown in Figure 9. Obviously, the KV shifting attention do not enhance the model's learning ability for this 3-gram tasks, but it also does not weaken the learning ability for this 3-gram tasks.

We must emphasize here that, the motivation of our KV shifting attention is to reduce the width and depth required for induction, we cannot expect KV shifting attention to greatly improve the memory ability of the model. In addition, n-gram tasks or some Markova data can actually be simulated with just one layer transformer with vanilla attention (Rajaraman et al., 2024). Therefore, the performance of Figure 9 is completely different from that of Figure 1a.

---

[14]In our experiment, if lr=2e-3 is used, Vanilla's performance will be quite poor (In Tr12-Te15, Vanilla will get approximately 0.5, while KV shifting will get approximately 0.7). For parameter tuning is not the most important thing we have at hand, we will try to find the most suitable hyper-parameters for each model in the future.

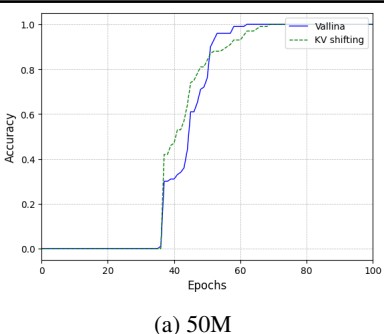
(a) 50M

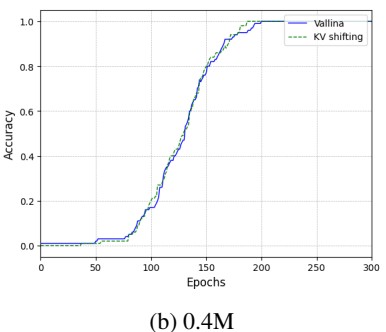
(b) 0.4M

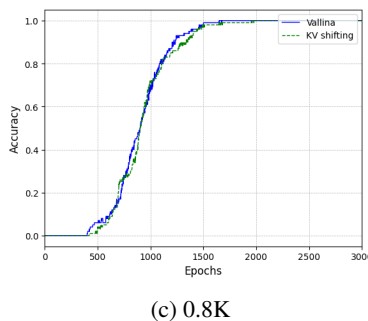
(c) 0.8K

Figure 9: Accuracy of learning 3-gram text using models of different sizes. In this experiments, there are 50M parameters model with 4 layers, 0.4M parameters model with 2 layers, 0.8K parameters model with 1 layer.

Another noteworthy thing is that intuitively, the shift operation on KV is beneficial for the fusion of local information, but the KV shift attention that integrates the previous token and current token does not contribute to the learning of 3-grams. This indicates that from the perspective of fusing local information, KV shift attention does not provide additional contributions, and perhaps position encoding is already sufficient. Perhaps it's because the bottleneck in learning n-grams is not in attention, but in other areas such as network width or the parameters numbers of MLP, which can be treated as a corollary of Rajaraman et al. (2024).

## F. Learnable parameter analysis

One worth studying is how these learnable parameters will change in a pre trained model. We conducted research on the 2.9B model, and due to our initialization, $\alpha_i$ and $\beta_i$ are independent random variables, but as training progressed, they became coupled with each other. As shown in the Table 5, we have counted the number of whether $\alpha_1 \leq \alpha_2$ and $\beta_1 \leq \beta_2$ in each KV pair.

Table 5: We calculate the number of whether $\alpha_1 \leq \alpha_2$ and $\beta_1 \leq \beta_2$ in each KV pair in 2.9B model with 500B token, the total numbers is 128.

|  | $\alpha_1 > \alpha_2$ | $\alpha_1 \leq \alpha_2$ |
|---|---|---|
| $\beta_1 > \beta_2$ | 50 | 17 |
| $\beta_1 \leq \beta_2$ | 9 | 52 |

As shown in the Table 5, we find that the numbers on the diagonal dominate, which means that KV pairs with same relative size relationships between $\alpha_1$ and $\alpha_2$, $\beta_1$ and $\beta_2$, has become the majority. And we find that the diagonal was already 47,43 when training 20B tokens. This indicates that for most heads, the model tends to use the key and value of the same token.

The heads in the upper right focus on the key of the $(i-1)^{th}$ token to obtain information about the $i^{th}$ token, while the heads in the lower left focus on the key of the $i^{th}$ token to obtain information about the $(i-1)^{th}$ token. And they are not symmetrical. We speculate that this is because the model can easily obtain information about the $(i-1)^{th}$ token by interacting with $i^{th}$ token under the causal mask, as the $(i-1)^{th}$ token token may contain some information about the $(i-1)^{th}$ token token to some extent. Therefore, the lower left corner will be relatively small.

### F.1. Different gate for shift

Besides, we find that for the trained 2.9B model with 500B tokens, $\sum_i \alpha_i$ and $\sum_i \alpha_i$ is away from 1, although they are 1 when initializing. However, a common gating mechanism is to use activation functions to control $\sum_i \alpha_i = 1$ and $\sum_i \alpha_i = 1$ during the training (e.g. $\alpha_1 = Sigmoid(a), \beta_1 = Sigmoid(b), \alpha_2 = 1 - \alpha_1, \beta_2 = 1 - \alpha_1$, where $a, b \in R$ is the learnable parameters, we call it KV shifting gate). In addition, during initialization, these parameters are all between 0 to 1, but as training progresses, some parameters become negative and far from zero.[15] Do we need to keep the model between 0 to 1

---

[15] e.g. $\alpha_1 = 0.08, \alpha_2 = 0.43, \beta_1 = 0.34, \beta_2 = -0.15$ in the $3^{th}$ KV pairs of the $17^{th}$ layers. The previous token's key to dominate the current token's key, while subtracting the previous token' value from the current token's value.

during the training process (e.g. $\alpha_i = \min(\max(\alpha_i, 0), 1)$, $\beta_i = \min(\max(\beta_i, 0), 1)$, we call it KV shifting 0 to 1.) ?

For this purpose, we conducted experiments on the 1.5B parameters model with 10B token, as shown in Figure 10a. Using more controls does not make the model learn better. Allowing $\alpha$ and $\beta$ to have a wider range of degrees of freedom may enable the model to learn richer features. For example, Elhage et al. (2021) discovered the presence of "anti-copying prefix-search" heads in vanilla model. Although we don't know what its function is, if we restrict $\beta_2 \geq 0$, it is likely to limit the generation of this kind of heads.

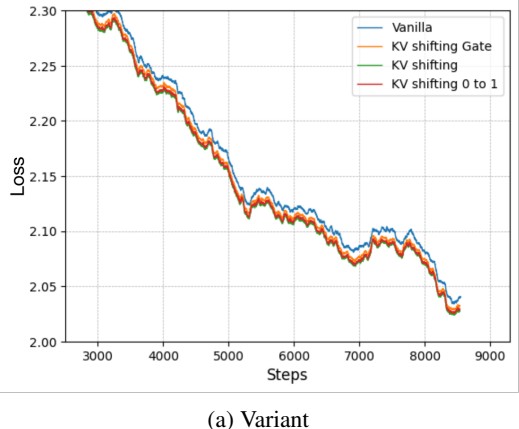

(a) Variant

Figure 10: Further experiments are conducted on a 1.5B model, where we trained 10B tokens.

## G. QKV shifting attention?

If we also shift Q in attention, it doesn't look like an ablation experiment, after all, it adds an extra part compared to the KV shifting attention. On the other hand, shifting Q is also far from our motivation. So we won't mention this section in the main text. But in order to better compare with some baseline methods which use the similar operation, such as Peng et al. (2023) which we discussed in the discussion section, they can shift all the information of Adjacent tokens. Another highly correlated works is Multi-DConv-Head Attention (So et al., 2021), they use architecture search and apply convolution to all Query, Key and Value. So we conduct shifting experiments on all Query, Key and Value, and the shifting is also per head.

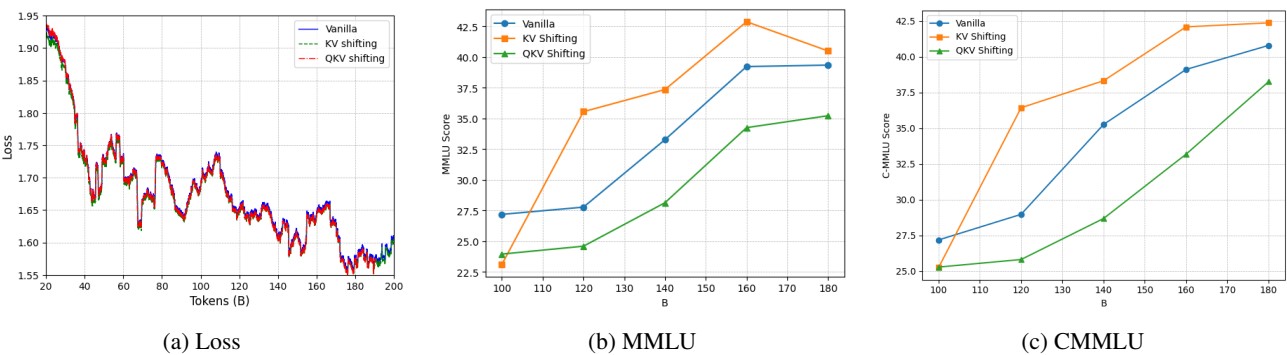

(a) Loss        (b) MMLU        (c) CMMLU

Figure 11: QKV shifting attention vs KV shifting attention in model with 19B parameters.

We trained a model for QKV shifting with 19B parameters. Due to some machine malfunctions, the last checkpoint of our QKV shifting attention is saved with 180B training tokens, and we don't plan to continue training until 200B. The loss curve is shown in Figure 11. From the loss curve, it seems that the two are similar. Then we evaluated the benchmarks as shown in Figure 11. It can be seen that the performance of shifting QKV is not as good as shifting KV, and even not as good as the vanilla. From the view of induction heads, the shifting of Q is difficult to contribute to the formation of it.

Finally, we will ask myself and answer a question when we recall Zhang et al. (2021). In their Figure 3, they try different position for token shift, such as "after Add", "before Norm", "after Norm", "before Add". From the perspective of our article, placing it in the place of K and V is the most suitable for forming induction heads. Shifting elsewhere can not alleviate the width requirement of induction heads.

# H. More comparison

There is an open source implements of KV shifting attention https://github.com/erogol/BlaGPT. If you want to verify the effectiveness of KV shifting attention, using their code and spending 8 NVIDIA-A100-80G plus 2 hours of training time is a good choice. They use a 0.2B model, and train with steps = 5100 and batch size = 0.5M. The training and validation set is sample from fineweb10B. And we use their code for the following comparison. Due to difficulty in compatibility with flash attention, out of memory (OOM) may occur on a larger scale for some method. So we use the code in https://github.com/microsoft/unilm/blob/master/Diff-Transformer/multihead_flashdiff_1.py for Differential Transformer (Ye et al., 2024a). And selective attention from https://github.com/fangyuan-ksgk/selective-attention-transformer/blob/main/model/model.py, although it remains OOM when context length is larger than 2048.

Note that all experiments here were conducted only once, and we can see that some numbers may be exceptions. However, in almost every learning rate and context length we experimented with, KV shifting attention achieved the best results.

Table 6: Vanilla model. Validation loss variation with learning rate and context length.

| Learning Rate | Context Length | | | | |
|---|---|---|---|---|---|
| | 256 | 512 | 1024 | 2048 | 4096 |
| 0.0001 | 3.7908 | 3.7078 | 3.6786 | 3.6758 | 3.6845 |
| 0.0002 | 3.6454 | 3.5839 | 3.5400 | 3.5308 | 3.5347 |
| 0.0004 | 3.5440 | 3.4558 | 3.4133 | 3.3778 | 3.3880 |
| 0.0010 | 3.4701 | 3.3816 | 3.3244 | 3.3082 | 3.2877 |
| 0.0020 | 3.4505 | 3.3605 | 3.3080 | 3.2813 | 3.2627 |
| 0.0040 | 3.4632 | 3.3701 | 3.3373 | 3.5104 | 3.2983 |

Table 7: KV shifting attention. Validation loss variation with learning rate and context length.

| Learning Rate | Context Length | | | | |
|---|---|---|---|---|---|
| | 256 | 512 | 1024 | 2048 | 4096 |
| 0.0001 | 3.7689 | 3.6968 | 3.6491 | 3.6402 | 3.6441 |
| 0.0002 | 3.6272 | 3.5616 | 3.5153 | 3.4975 | 3.4929 |
| 0.0004 | 3.5308 | 3.4440 | 3.3903 | 3.3686 | 3.3692 |
| 0.0010 | 3.4603 | 3.3691 | 3.3166 | 3.2838 | 3.2770 |
| 0.0020 | 3.4381 | 3.3686 | 3.2983 | 3.2618 | 3.2404 |
| 0.0040 | 3.4766 | 3.4327 | 3.3097 | 3.3109 | 3.3017 |

Table 8: QKV shifting attention. Validation loss variation with learning rate and context length.

| Learning Rate | Context Length | | | | |
|---|---|---|---|---|---|
| | 256 | 512 | 1024 | 2048 | 4096 |
| 0.0001 | 5.8747 | 3.7215 | 3.6750 | 3.6752 | 3.6798 |
| 0.0002 | 3.6543 | 3.5793 | 3.5301 | 3.5391 | 3.5213 |
| 0.0004 | 3.5482 | 3.4584 | 3.4190 | 3.3847 | 3.3836 |
| 0.0010 | 3.4678 | 3.3783 | 3.3307 | 3.3013 | 3.2835 |
| 0.0020 | 3.4518 | 3.3769 | 3.3081 | 3.2725 | 3.2620 |
| 0.0040 | 3.4994 | 3.3710 | 3.3063 | 3.3486 | 3.2662 |

Table 9: K shifting attention. Validation loss variation with learning rate and context length.

| Learning Rate | Context Length | | | | |
|---|---|---|---|---|---|
| | 256 | 512 | 1024 | 2048 | 4096 |
| 0.0001 | 3.7787 | 3.7109 | 3.6567 | 3.6430 | 3.6667 |
| 0.0002 | 3.6519 | 3.5758 | 3.5356 | 3.5110 | 3.5085 |
| 0.0004 | 3.5471 | 3.4541 | 3.4037 | 3.3820 | 3.3718 |
| 0.0010 | 3.4753 | 3.3861 | 3.3240 | 3.2976 | 3.2791 |
| 0.0020 | 3.4531 | 3.3570 | 3.3038 | 3.2733 | 3.2564 |
| 0.0040 | 3.4993 | 3.3699 | 3.3080 | 3.3768 | 3.2795 |

Table 10: V shifting attention. Validation loss variation with learning rate and context length.

| Learning Rate | Context Length | | | | |
|---|---|---|---|---|---|
| | 256 | 512 | 1024 | 2048 | 4096 |
| 0.0001 | 3.7742 | 3.7061 | 3.6698 | 3.6512 | 3.6610 |
| 0.0002 | 3.6366 | 3.5574 | 3.5253 | 3.5044 | 3.5038 |
| 0.0004 | 3.5300 | 3.4494 | 3.3980 | 3.3775 | 3.3639 |
| 0.0010 | 3.4599 | 3.3711 | 3.3234 | 3.2895 | 3.2745 |
| 0.0020 | 3.4425 | 3.3489 | 3.2915 | 3.2622 | 3.2658 |
| 0.0040 | 3.4805 | 3.3556 | 3.3798 | 3.3130 | 3.4101 |

Table 11: Differential Transformer (Ye et al., 2024a). Validation loss variation with learning rate and context length.

| Learning Rate | Context Length | | | | |
|---|---|---|---|---|---|
| | 256 | 512 | 1024 | 2048 | 4096 |
| 0.0001 | 3.9601 | 3.9023 | 3.8693 | 3.8336 | 3.8279 |
| 0.0002 | 3.7360 | 3.6692 | 3.6280 | 3.6149 | 3.6123 |
| 0.0004 | 3.5909 | 3.5191 | 3.4705 | 3.4462 | 3.4362 |
| 0.0010 | 5.8422 | 3.4158 | 3.3718 | 3.3356 | 3.3648 |
| 0.0020 | 3.7052 | 3.7530 | 3.7660 | 4.1150 | 3.3187 |
| 0.0040 | 7.8280 | 6.6233 | 6.4263 | 3.3948 | 3.4779 |

Table 12: Differential Transformer (Ye et al., 2024a). Validation loss variation with learning rate and context length. We enhance Differential Transformer with query and key normalization(Henry et al., 2020), which is the base setting of https://github.com/erogol/BlaGPT.

| Learning Rate | Context Length | | | | |
|---|---|---|---|---|---|
| | 256 | 512 | 1024 | 2048 | 4096 |
| 0.0001 | 3.9622 | 3.8999 | 3.9122 | 3.8484 | 3.8559 |
| 0.0002 | 3.7218 | 3.6624 | 3.6245 | 3.6154 | 3.6222 |
| 0.0004 | 3.5800 | 3.5080 | 3.4703 | 3.4688 | 3.4476 |
| 0.0010 | 3.5305 | 3.4076 | 3.3840 | 3.3512 | 3.3482 |
| 0.0020 | 3.4998 | 3.3930 | 3.3478 | 3.3635 | 3.3543 |
| 0.0040 | 3.5607 | 3.4142 | 3.3566 | 3.3573 | 3.3489 |

Table 13: Selective Attention (Leviathan et al., 2024) Validation loss variation with learning rate and context length. Vanilla model's peak memory consumption is 42584 MiB when context length is 4096, KV shifting attention is 43360 MiB. The peak memory consumption of selective attention is 69500 MiB when context length is 1024. We will supplement this part of the experiment when selective attention is implemented more efficiently.

| Learning Rate | Context Length | | | | |
|---|---|---|---|---|---|
| | 256 | 512 | 1024 | 2048 | 4096 |
| 0.0001 | 3.7922 | 3.7109 | 3.6817 | OOM | OOM |
| 0.0002 | 3.6580 | 3.5842 | 3.5466 | OOM | OOM |
| 0.0004 | 3.5538 | 3.4744 | 3.4240 | OOM | OOM |
| 0.0010 | 3.4768 | 3.3985 | 3.3594 | OOM | OOM |
| 0.0020 | 3.4575 | 3.4100 | 3.3613 | OOM | OOM |
| 0.0040 | 3.4850 | 3.4618 | 3.4128 | OOM | OOM |

## I. Metrics for evaluation

In this section, we conducted MMLU evaluation under three condition, few shot (5 - shot), zero shot and cloze (zero shot). The setting of cloze is followed by Waleffe et al. (2024), which intends to break away from the format of standard multiple-choice and directly measure the knowledge. The test results are shown in Table 14. It can be seen that compared to vanilla, KV shifting attention not only enhances the ability of in-context learning, but also accelerates the model's learning of world knowledge. Moreover, the format of multiple-choice questions may be advantageous for KV shifting attention. The model can easily use context to compare the possibilities of various options and select the option with the highest probability.

Table 14: We compare vanilla and KV shifting at 2.9B model with 500B training tokens by using different evaluation metric.

| Benchmark | Vanilla | | | KV shifting | | |
|---|---|---|---|---|---|---|
| | Cloze | Zero | Few | Cloze | Zero | Few |
| MMLU | 30.41 | 33.14 | 37.26 | 32.17 | 37.13 | 40.88 |

In addition, under different evaluation metric, KV shifting attention achieved better results, which reflects the robust of KV shifting attention.

## J. Compatibility with multi latent attention

Recently, a highly efficient attention method for reasoning, namely multi latent attention (MLA), has achieved good results (Liu et al., 2024). However, due to the fact that the attention calculation of MLA in the inference stage is at the latent level, it is difficult to be compatible with some improvements to MLA, such as QK normalization (Henry et al., 2020), Rotary position embedding (RoPE) Su et al. (2024). So much so that their roped vectors were shared across all the heads. But KV shift attention can be compatible in this way, that is, we ensure that the key of some the heads in each layer are generated entirely by the previous token, and the value of some the heads in each layer are generated entirely by the previous token.

On the other hand, what is the necessity of performing KV Shift on MLA? According to our analysis of attention execution induced heads in Section 3, the width of the model constrains its ability to learn and represent induced heads, while the projection operation of MLA brings the bottleneck of width to the size of the latent space.

We conducted the following experiment with the same basic setup as H, for example, the size of hidden states is 768, with a total of 12 heads. We did not perform low rank projection on query. For MLA, we perform low rank projection on key and value to 256 dimensions, and then the dimension with RoPE in key and value is 32, the dimension without RoPE is 32, and the dimension of query is 64. Due to the total number of parameters being about 5% more than in Appendix H, we do not further align the total parameter numbers. For MLA with KV shifting, we shift the key of one head, and the value of another head. We find that lr=2e-3 is the best for both of them. The results are shown below:

We found that as the sequence length increases, the overall performance of shift KV is also better than MLA. This also indicates the enhancement of the model's ability to utilize context.

Table 15: Enhance MLA with KV shifting.

| Method | Validation loss | | | | |
| --- | --- | --- | --- | --- | --- |
| | 256 | 512 | 1024 | 2048 | 4096 |
| MLA | 3.5098 | 3.415 | 3.3608 | 3.3188 | 3.2976 |
| MLA w KV shifting | 3.4987 | 3.4069 | 3.3414 | 3.3002 | 3.2753 |

## K. Python script for generating induction data

This is a Python script that generates induction data. The model will randomly select a number from mid val to max val as a token. If this token has already appeared, the next token will be the same as the previous one. When the sequence length is less than 512, 0 will be added. If there is no induction data in the sequence, it will be regenerated. The predict positions returned by this function are the positions where the accuracy of the induction is calculated.

```python
def generate_array(length=512, min_val=1, mid_val=10,max_val=8000):
    lis = list(range(mid_val+1,max_val))
    random.shuffle(lis)
    lis = lis[:length]
    array = []
    predict_positions = []
    di = {}
    while len(array)<length:
        x = random.choice(lis)
        if (len(array) == 0) or (x!=array[-1]):
            if x not in di:
                if len(array)>0:
                    di[array[-1]] = x
                di[x] = -1
                array.append(x)
            else:
                predict_positions.append(len(array))
                array.append(x)
                array.append(di[x])
                return array + [0]*(512-len(array)), predict_positions[0:1]
    if len(predict_positions) == 0:
        return generate_array(length, min_val, mid_val,max_val)
```

## L. Python code for drawing a streamline diagram

We provide a Python code for drawing streamline diagrams in Figure 2.

```python
import torch
import numpy as np
import matplotlib.pyplot as plt
def f(alpha,beta,ot = 0):
    alpha1 = alpha
    alpha2 = 1 - alpha
    beta1 = beta
    beta2 = 1 - beta
    a1 = torch.exp(alpha1)/(2*torch.exp(alpha1)+torch.exp(alpha2)+ot)
    a2 = torch.exp(alpha2)/(2*torch.exp(alpha1)+torch.exp(alpha2)+ot)
    S = (2*torch.exp(alpha1)+torch.exp(alpha2)+ot)
    target = a2*beta1 + (beta2/S)
    no1 = (beta1/S) + beta2*a2
    no2 = 2*a1*beta1 + a2+beta2

    exp_target = torch.exp(target)
    exp_no1 = torch.exp(no1)
    exp_no2 = torch.exp(no2)
    exp_ot = torch.exp(ot)
    denominator = exp_target + 2 * exp_no1 + exp_no2 + exp_ot
    prob_ratio = exp_target / denominator
    loss = -torch.log(prob_ratio)
    return loss

X,Y = np.meshgrid(np.linspace(0, 1, 1000), np.linspace(0, 1, 1000))
X = torch.tensor(X, requires_grad=True)
Y = torch.tensor(Y, requires_grad=True)
Z = f(X,Y,0.00001,0)
Z.sum().backward()

grad_X = X.grad
grad_Y = Y.grad
fig, ax = plt.subplots()
strm = ax.streamplot(
    X.detach().numpy(), Y.detach().numpy(),
    -grad_X.detach().numpy(), -grad_Y.detach().numpy()
)
levels = np.arange(0, 10, 0.05)
contours = plt.contour(
    X.detach().numpy(), Y.detach().numpy(),
    Z.detach().numpy(), levels=levels, colors='black'
)
ax.set_xlabel(r'$\alpha_1$')
ax.set_ylabel(r'$\beta_1$')
plt.show()
```

# M. Pytorch code for KV shifting attention

We provide the following Python code that can easily implement KV shifting attention with rotary embedding. In this example, we used convolution operation to perform shifting operations.

```python
from torch import nn
import torch.nn.functional as F
from flash_attn import flash_attn_func
def custom_convolution(U, K):
    bs, seq, h, d = U.shape
    h, w = K.shape
    padding = (w - 1, 0)
    U_padded = F.pad(U, (0, 0, 0, 0, *padding))  #  (bs, seq+w-1, h, d)
    U_unfolded = U_padded.unfold(1, w, 1)  # (bs, seq+w-1, h, d, w)
    K_expanded = K.unsqueeze(0).unsqueeze(0).unsqueeze(-2)  #  (1, 1, h, 1, w)
    V_unfolded = U_unfolded * K_expanded  #  (bs, seq, h, d, w)
    V = V_unfolded.sum(dim=-1)  #  (bs, seq, h, d)
    return V
def __init__(self):
    K = torch.rand(self.num_kv_heads,1)
    V = torch.rand(self.num_kv_heads,1)
    self.K = nn.Parameter(torch.cat([K,1-K],dim=1))
    self.V = nn.Parameter(torch.cat([V,1-V],dim=1))
def foward(self,q,k,v):
    k = custom_convolution(k, self.K)
    v = custom_convolution(v, self.V)
    q, k= self.rotary_emb(q, k, seqlen_offset=0)
    attn_outputs = flash_attn_func(
                    q,
                    k,
                    v,
                    causal=True
                )
```

The following code can be used for inference:

```python
if past_key_value is None:
    self.last_k,k  = k[:,-1:],custom_convolution(k, self.K)
    self.last_v,v= v[:,-1:],custom_convolution(v, self.V)
else:
    self.last_k,k = k, self.K[:,:1]*self.last_k + self.K[:,1:]*k
    self.last_v,v = v, self.V[:,:1]*self.last_v + self.V[:,1:]*v
```

# N. Experimental setup details

We conducted toy models for induction heads on 8 Nvidia A100-80G GPUs, with 512 samples per GPU. The learning rate is $2e-4$ with 1000 steps warm-up. For large language model, due to privacy reasons, we are unable to provide a detailed description of the training data here. Our training data contains a large amount data from Fineweb-edu (Lozhkov et al., 2024), as well as some other filtered data. We have listed the parameters of our models of various sizes below, some of which have also been mentioned in the main text. We used a larger RoPE's base here because previous study (Xu et al., 2024) has shown that the longer the context length, the larger the base required, while the default base=10,000 is relatively small, even for 2048 windows. Therefore, a relatively large value has been uniformly set here.

Table 16: Configuration.

| PARAMETERS | 1.5B | 2.9B | 6.7B | 13B | 19B |
|---|---|---|---|---|---|
| HIDDEN SIZE | 2,048 | 2,560 | 4,096 | 5,120 | 6,144 |
| LAYERS | 28 | 32 | 32 | 40 | 48 |
| HEAD NUMBER | 16 | 20 | 32 | 40 | 48 |
| KV NUMBER | 16 | 4 | 32 | 40 | 4 |
| FFN SIZE | 5,504 | 8,704 | 11,008 | 13,824 | 16,384 |
| MAX LENGTH | 2,048 | 4,096 | 2048 | 2,048 | 12,288 |
| TOTAL TOKENS | 10B | 500B | 10B | 10B | 200B |
| VOCAB SIZE | 36,000 | 48,000 | 36,000 | 36,000 | 48,000 |
| GPU | A100-80G | H800-80G | A100-80G | A100-80G | A800-80G |
| GPU NUMBERS | 64 | 512 | 64 | 128 | 240 |
| CONTEXT LENGTH | 2,048 | 4,096 | 2,048 | 2,048 | 12,288 |
| BATCH SIZE PER GPU | 8 | 8 | 8 | 8 | 1 |
| LEARNING RATE | 2E-4 | 8E-4 | 2E-4 | 2E-4 | 2E-4 |
| LEARNING RATE SHEDULE | | | CONSTANT | | |
| WARM-UP STEPS | 1,000 | 600 | 1,000 | 1,000 | 3,000 |
| OPTIMIZER | | ADAM WITH $\alpha_1 = 0.9, \alpha_2 = 0.95$, WEIGHT DECAY=0.1 | | | |
| ROPE'S BASE | | | 100,000 | | |

