# OpenReview forum: "KV Shifting Attention Enhances Language Modeling"
_ICML.cc/2025/Conference — ICML 2025 poster_

### Official Review · Reviewer_SEX2 · 2025-03-13

**Overall Recommendation:** 4

**Summary:**

- This paper proposes a new attention mechanism named KV shifting attention to enhance the in context learning ability of transformer.
- The paper aims at enhance inductive head bias of transformer by shifting the key&value vectors.
- This paper provides extensive analysis to show the effectiveness of the KV shifting attention in enhancing learning ability and reduce model width requirements,.

**Claims And Evidence:**

- The paper clearly written and well motivated.
- The proposed method is simple and elegant, by only introducing few parameters and computation.
- Yes, extensive experiments are provided to support the claims.

**Essential References Not Discussed:**

None

**Experimental Designs Or Analyses:**

Yes, Extensive experiments have been conducted to validate the effectiveness of the method.

**Methods And Evaluation Criteria:**

Yes

**Other Comments Or Suggestions:**

None

**Other Strengths And Weaknesses:**

weakness:
- As claimed in line 398, the key to the effectiveness is to achieve better induction heads by decoupling key-value pairs in time sequence. I wonder how 1d causal conv performs compared with kv shifting. In addition, the novelty of the proposed might be weak as it is a simplified version of short conv.
- Training and inference speed comparison can be included.

**Questions For Authors:**

None

**Relation To Broader Scientific Literature:**

The proposed method is simple and seems effective, it could be included in future work and more scaled-up models to enhance the language modeling ability.

**Theoretical Claims:**

Yes.
The authors aim at formulating the behavior of induction heads in one or two transformer layers,

---

> ### Author Rebuttal · Authors · 2025-03-30
>
> Thank you very much for your review. I hope the following response can address your concerns.
>
> **Causal Conv**
>
> Thank you very much for pointing this out. As demonstrated in our discussion, the shifting operation can be seen as a short convolution with kernel size 2.
>
> (a) From an operational perspective, this is not a very new operation. But where to perform this operation is crucial. As shown in Figure 11, using short convolutions on key and value to decouple them in time is beneficial for the emergence of benchmarks such as MMLU. However, if a similar operation is also used on query, interestingly, the effect will actually deteriorate.
>
> (b) On the other hand, I think we have clarified to some extent the true role of short convolutions in attention. Decoupling the key and value sequences to achieve better context learning. Some previous works believed that this convolution was intended to enhance local information exchange.
>
>
> **Training and inference speed**
>
> We can note that the computational cost of the kv shift operation is $O(nd)$, Compared to $O (nd^2)$ for matrix projection and $O(n^2d)$ for attention calculation, this is much small.  At the same time, under group query attention, the cost of this part is even smaller. Therefore, there was not much additional computational cost added during the training and inference phases. When considering inference, the additional storage space required is $O (d)$ to store the previous time's kv, which has a relatively small storage overhead compared to the original $O(nd)$. Therefore, kv shift does not have significant memory access overhead during the inference phase. The actual end-to-end training and inference speed is related to the framework used. In our own framework, the additional time cost is less than 2% for a 3B model with kv shift.

---

### Official Review · Reviewer_fNW3 · 2025-03-13

**Overall Recommendation:** 4

**Summary:**

This paper introduces KV shifting attention, a modification to the standard transformer attention mechanism that changes how keys and values are processed. By decoupling the temporal relationship between keys and values, the model can more efficiently learn induction heads - a critical mechanism for in-context learning in language models. Hence, this could suggest a new approach to improve language modeling.

The authors provide theoretical analysis showing that KV shifting attention can reduce the width and depth requirements for forming induction heads compared to vanilla transformers. While traditional transformers need at least two layers to implement induction heads effectively, the authors prove that KV shifting attention can implement this mechanism with a single layer.

The authors evaluate their approach on both small-scale experiments designed to test induction capabilities and large-scale language model pre-training (models ranging from 1.5B to 19B parameters). Their experiments demonstrate that models with KV shifting attention consistently outperform vanilla attention baselines across model scales, with improved performance on standard benchmarks and faster convergence during training.

**Claims And Evidence:**

There are four main claims in the paper, each of which are well-supported.

- Improved theoretical foundation for induction heads: The authors provide formal proofs (Theorems 3.2 and 3.3) that KV shifting attention requires less model complexity to form induction heads.
- Better induction capabilities: Experimental results demonstrate that KV shifting attention models can learn induction patterns more efficiently than vanilla attention models of equivalent or even larger size.
- Enhanced language modeling performance: The comprehensive evaluations on different models sizes show consistent improvements in both convergence speed and final performance across datasets and benchmarks.
- Robustness to hyperparameters: The experiments with different random seeds, learning rates, and model sizes provide evidence that the improvements are consistent.

**Essential References Not Discussed:**

The following paper might be highly relevant: PermuteFormer: Efficient Relative Position Encoding for Long Sequences.

**Experimental Designs Or Analyses:**

- The toy model experiments isolate the induction capabilities being studied, allowing clear observation of the learning dynamics.
- The scaling experiments with models from 1.5B to 19B parameters follow best practices in the field and allow for meaningful comparisons.
- The ablation studies (Section 4.5) help identify which components of the proposed method contribute to the performance improvements.
- The robustness experiments with different random seeds and learning rates strengthen confidence in the results.

**Methods And Evaluation Criteria:**

- The authors evaluate on a suite of standard language modeling benchmarks.
- The scaling experiments compare performance across model sizes and training compute.
- Experiments with different hyperparameters and random seeds verify the robustness of the improvements.
- The toy task suggested measure the specific capability the method aims to improve.

**Other Comments Or Suggestions:**

It would be very interesting to see how this method would interact with more recently suggested transformer architectures, or with other techniques such as prompting.

**Other Strengths And Weaknesses:**

Strengths:
- The paper presents a simple yet effective modification to the attention mechanism with minimal computational overhead.
- The theoretical analysis provides clear insights into why the method works.
- The comprehensive evaluation across model scales demonstrates practical value for real-world language modeling.
- The method is compatible with existing transformer optimization techniques (as shown in Appendix J).

Weaknesses:
- The paper could provide more analysis of what types of tasks or text patterns benefit most from the improved induction capabilities.
- While the authors show that KV shifting attention helps with math reasoning (Appendix E.2), a more detailed analysis of reasoning capabilities would strengthen the paper.
- The paper mentions but does not extensively explore potential limitations of the approach, such as whether there are specific scenarios where vanilla attention might be preferable.

**Questions For Authors:**

- Would this method also apply for encoder-decoder structures? Would there need to be some modifications in the mechanism?
- Would this method also apply when training with multimodal models, or have benefit when adapting the two modalities?

**Relation To Broader Scientific Literature:**

- It builds on the mechanistic interpretability work by Elhage et al. (2021) and Olsson et al. (2022) who identified and characterized induction heads.
- The theoretical analysis extends recent work on the theoretical capabilities of transformers, such as Wang et al. (2024) and Sanford et al. (2024a, 2024b).
- The approach relates to other architectural modifications in language modeling, such as RWKV (Peng et al., 2023), Mamba (Gu & Dao, 2023), and RetNet (Sun et al., 2023), but focuses specifically on enhancing the standard transformer architecture.
- The paper connects to broader work on in-context learning mechanisms in large language models, an active area of research.

**Theoretical Claims:**

The theoretical claims provided in Section 3 are supported by formal proofs.

- Theorem 3.2 states that a two-layer vanilla transformer with specific parameter settings can approximate the induction heads mechanism with a certain error bound.
- Theorem 3.3 states that a one-layer KV shifting attention model can exactly implement the induction heads mechanism.
- Theorem 3.4 provides an analysis of the learning dynamics for KV shifting attention.

The proofs appear sound, with detailed derivations provided in the appendices.

---

> ### Author Rebuttal · Authors · 2025-03-31
>
> Thank you very much for your meticulous review. We hope our response can be helpful to you.
>
> **More analysis**
>
> (a) **What types of tasks or text patterns benefit most from the improved induction capabilities?** To be honest, it is difficult to exhaust what kind of pattern is more suitable, but we can provide some examples about what kind of pattern is suitable and what kind of pattern cannot be gained. As we have stated in our paper, hop k tasks can achieve better results, and math tasks can also achieve better results. However, n-gram tasks do not have very good results. We later conducted experiments on context free gramma(CFG) [1]. We found that KV shifting attention **does not** improve the learning of CFG, the convergence speed and final performace are almost the same as the vanilla model. Therefore, we speculate that the enhancement of language modeling by kv shifting comes from semantics rather than context free syntax.
>
> (b) **A more detailed analysis of reasoning capabilities.**
>
> The analysis in this section might also serve as part of the answer to the previous question. Based on our test of mathematical ability, we find that KV shifting attention can effectively package the information of adjacent tokens and store them separately in key and value for ease of subsequent operations. For example, "3+5=", the information of "3" is used as the value, shifted to "+", and the value of "5" is shifted to "=". Then, "=" uses two attention heads, the first one obtains the information of "3+" by following "+", and the other head obtains the information of "5" by following "=", and then uses MLP to obtain the correct answer "8".
>
> On the other hand, induction heads are also more conducive to helping establish relationships between things. For example, if a sentence contains "Beijing roast duck" or "Beijing, China", then kv shifting can easily create some key value pairs in one layer of attention, such as<key: Beijing, value: Roast Duck><key: Beijing, value： In the following text, it is easy to associate Beijing with China or roast duck, which is beneficial for the model to answer questions such as which country Beijing is in or what its cuisine is.
>
> (c) **In terms of potential limitations.** We speculate that KV shifting may enhance the pattern of repetition in the model due to the enhancement of induction heads. Although there is no conclusive evidence, I would like to share an interesting discovery. We later trained a model with 14B parameters with 15T tokens. On the  benchmark math500, we find that the optimal generation configuration is to set the repetition penalty to 1.1, but for qwen2.5-14b, only the repetition penalty needs to be set to 1.05.
>
> **More extensive applications**
>
> (a) **For encoder-decoder structures.**  Due to the absence of a causal mask, the direction of shift in the encoder can be either forward or backward. I believe that KV shift can also enhance the encoder decoder model, as it can benefit from easier to implement induction mechanisms.
>
> (b) **Multimodal model.**  In visual models, there is a similar approach to using token shift [2], which they believe is beneficial for "back and forth for motion captioning". Therefore, I believe that similar operations can contribute to multimodal models. If it can really work, then the real reason for its effectiveness will be even more interesting. Does it enhance local information exchange, or did it enhance something similar to the induction heads mechanism?
>
> [1] Physics of language models: Part 1, learning hierarchical language structures.
>
> [2] Token ShiftTransformer for Video Classification.

---

> > ### Comment · Reviewer_fNW3 · 2025-04-09
> >
> > I thank the authors for their rebuttal. I have enjoyed reading the paper and the rebuttal. While I think that this line of work is an important one, I think the reviewer fHVj raises a valid concern. Hence I retain my score

---

### Official Review · Reviewer_fHVj · 2025-03-14

**Overall Recommendation:** 2

**Summary:**

Based on the analysis of 2-layer attention, the authors found that 2-layer attention cannot effectively represent information flow from i+1 -> k -> j (for j>=i, k<i). Therefore, the authors proposed KV shifting attention to address this issue. The authors also conducted large-scale pre-training experiments to validate its effect.

## update after rebuttal
Regarding my main question, "whether the limitation proposed in this paper still exists when attention layers >=3", the authors mainly use Figure 8 to respond. Figure 8 does provide some explanation, but I believe it's insufficient and indirect. Therefore, I maintain my score.

**Claims And Evidence:**

1. I'm not entirely sure if my understanding is correct. According to Property 1, for j to indirectly attend to i+1 through token k, k needs to be k>= i+1. This is indeed a limitation of attention. However, this property only considers the 2-layer case. If LLMs have more than 2 layers, would this problem be solved? I think this is similar to MLP's expressiveness: although 2-layer MLPs cannot represent non-linearly separable functions, 3-layer MLPs can represent all functions. Therefore, if the limitations proposed in Property 1 can be solved by 3-layer attentions, is this problem still serious considering that current LLMs generally have dozens of attention layers?
2. Following point 1, the impact of model depth on Property 1 and experimental results has not been verified.
3. Following point 1, according to figure 3, the impact of KV shifting on loss is minor, especially in the 19B model. Does this confirm the limitation of this paper's motivation: for larger and deeper LLMs, the limitations of 2-layer attention revealed by property 1 disappear or are no longer a key issue?

**Essential References Not Discussed:**

No

**Experimental Designs Or Analyses:**

Please see the third point in Claims And Evidence.

**Methods And Evaluation Criteria:**

Yes. However, according to Figure 3, its effect is not significant.

**Other Comments Or Suggestions:**

Typos: line 84, Property 3.2 -> Property 1
Citation: Press,O., lack of year information.

**Other Strengths And Weaknesses:**

No

**Questions For Authors:**

Please refer to Claims And Evidence.

**Relation To Broader Scientific Literature:**

I think this is similar to MLP's expressiveness: although 2-layer MLPs cannot represent non-linearly separable functions, 3-layer MLPs can represent all functions.

**Theoretical Claims:**

I tried to understand property 1. Please see Claims And Evidence regarding my concerns.

---

> ### Author Rebuttal · Authors · 2025-03-30
>
> Thank you very much for your valuable review comments. I hope the following response can answer your concerns.
>
> **Regarding expressive power**
>
> Firstly, if I understand correctly, the "three" in the three-layer MLP you mentioned refers to the layer meaning of neuron nodes, which actually corresponds to MLP with two layers of parameter. I think using more layers can also achieve the function of induction heads, as shown in Figure 1 (a), four layers of attention can also achieve the function of induction heads. But for the simple task of induction heads, 4 layers cannot improve compared to 2 layers, so more difficult tasks can be considered. For example, you can refer to the hop k task in Appendix E1, where KV shifting attention can achieve better performance in hop k tasks. Here, we compared L=2,3,4,5,6 layers.  In terms of motivation, the two layers situation has inspired the work, and rigorous theoretical analysis of multiple layers is difficult, which we leave for the future.
>
> **The impact of model depth**
>
> You can refer to the hop k task in Appendix E1, where KV shifting attention can achieve better performance in hop k tasks. Here, we compared L=2,3,4,5,6 layers. From the experimental results, we speculate that multi-layer did not alleviate this situation. Because if there is relief, multi-layer attention can better achieve multi hop through some shortcut, thereby filling the gap between Figure 8 (b) and Figure 8 (c). In fact, this gap did not decrease with the increase of layers. I think the causal mask has caused such restrictions on information flow.
>
> **Loss and Benchmark**
>
> (a) Firstly, the learning rate of the model trained on the 19B is 2e-4, which is relatively small. As shown in our Figure 4 at a scale of 1.5B, the gap between the two will narrow in the case of primary school learning rate. If a larger learning rate is adopted, it may be speculated that the gap in loss between the two will increase. (b) Secondly, I would like to point out that loss can be confusing during the pre training process of large language models, because there are a large number of tokens that can be easily predicted using n-gram information, without truly utilizing contextual information[1]. Therefore, we compared some benchmarks, among which benchmarks like MMLU are relatively reliable[2], as shown in Figure 11, where KV shifting attention achieved good results on MMLU. (c) The results of some toy tasks in Appendix make us believe that the model can also achieve better in context learning ability at larger scales.
>
>
> [1]This is particularly evident in the training of long texts, where models with similar losses have significantly different abilities for long-distance information retrieval. "Can Perplexity Reflect Large Language Model's Ability in Long Text Understanding?"
>
> [2] For example, we can observe that the performance on MMLU can clearly separate transformer  models from mamba model in Table 4 of "An Empirical Study of Mamba-based Language Models".

---

### Decision · Program_Chairs · 2025-05-01

**Decision:**

Accept (poster)

**Comment:**

I recommend accepting this paper. The KV shifting attention mechanism provides a theoretically inspired approach to improve transformer induction capabilities. Two reviewers gave accept ratings, noting the method's simplicity, theoretical foundation, and evaluation across model scales. One reviewer gave a weak reject, the question is around its relevance in deeper models, but the authors' rebuttal offers more evidence showing benefits across different layer depths. The improvements in convergence speed and performance at various scales, combined with minimal computational overhead, make this a solid contribution to the community.